# Multi-modal Queried Object Detection in the Wild

**Yifan Xu**[1,3†,*] **Mengdan Zhang**[2†]**, Chaoyou Fu**[2]**,**
**Peixian Chen**[2]**, Xiaoshan Yang**[1,3,4]**, Ke Li**[2]**, Changsheng Xu**[1,3,4‡]
[1]MAIS, Institute of Automation, Chinese Academy of Sciences   [2]Tencent Youtu Lab
[3]School of Artificial Intelligence, University of the Chinese Academy of Sciences
[4]Peng Cheng Laboratory
{yifan.xu, csxu}@nlpr.ia.ac.cn, davinazhang@tencent.com

## Abstract

We introduce MQ-Det, an efficient architecture and pre-training strategy design to utilize both textual description with open-set generalization and visual exemplars with rich description granularity as category queries, namely, **M**ulti-modal **Q**ueried object **Det**ection, for real-world detection with both open-vocabulary categories and various granularity. MQ-Det incorporates vision queries into existing well-established language-queried-only detectors. A plug-and-play gated class-scalable perceiver module upon the frozen detector is proposed to augment category text with class-wise visual information. To address the learning inertia problem brought by the frozen detector, a vision conditioned masked language prediction strategy is proposed. MQ-Det's simple yet effective architecture and training strategy design is compatible with most language-queried object detectors, thus yielding versatile applications. Experimental results demonstrate that multi-modal queries largely boost open-world detection. For instance, MQ-Det significantly improves the state-of-the-art open-set detector GLIP by +7.8% AP on the LVIS benchmark via multi-modal queries without any downstream finetuning, and averagely +6.3% AP on 13 few-shot downstream tasks, with merely additional 3% modulating time required by GLIP. Code is available at https://github.com/YifanXu74/MQ-Det.

## 1   Introduction

As the thriving of large-scale vision-language pre-training models [25, 48, 24, 29, 27, 44, 20], object detection recently has ushered in a new fashionable paradigm, which locates the desired objects via a queried text. Benefited from the generalization of the pre-training models on large scale data [26, 13, 36, 43, 22, 18, 37, 31, 35], such a text queried paradigm makes steady progress on the long road to open-set object detection.

Compared with previous fixed category sets (usually represented by finite numbers), the foregoing text query has the merit of representing broad concepts, but also has an intrinsic limitation of insufficient description granularity [4, 9, 32]. For example, class homonyms (*e.g.*, "bat" can be a piece of wood or a kind of animal) lead to ambiguous queries. Meanwhile, as exemplified in Fig. 1, for fine-grained fish species detection, it is usually struggled to use limited text to describe the fish with specific patterns. Empirically, one straightforward solution for the insufficient description granularity problem is to design additional textual description, but there are three distinct obstacles: 1) it is difficult to comprehensively describe visual details [52], and constructing textual description for a large amount of categories is a laborious work. 2) The longer queried text increases the understanding difficulty of the pre-training model and 3) brings more computational overhead. Experimentally, the state-of-the-art (SoTA) text queried detector GLIP [25] merely improves average precision (AP) from

---

*Work done when interning at Tencent Youtu Lab. †Equal contribution. ‡Corresponding author.

37th Conference on Neural Information Processing Systems (NeurIPS 2023).

17.7% to 18.4% on the Aquarium dataset [23] (a fine-grained fish species detection dataset), even with extra manually designed textual description for some categories.

A picture paints a thousand word. Compared with the text, the image can provide richer clues about the target object. But at the same time, the human-generated text has higher information density and thereby brings stronger generalization capability [32, 8, 2]. In light of this, a natural idea springs up, *i.e.*, combining the text and the image to constitute a multi-modal query, which has the advantages of both breadth of the former and rich granularity of the latter. Nevertheless, how to acquire such a multi-modal queried detection model still faces challenges: 1) directly finetuning with limited visual exemplars, typically in previous vision-queried few-shot detection methods [49, 39, 21, 16], leads to the risk of catastrophic forgetting. 2) Large foundation models have good generalization but require heavy training burden if re-organized and trained from scratch (e.g., over 30 million data storage and almost 480 V100 GPU days for GLIP [25, 48]).

This work fills in the blank of **M**ulti-modal **Q**ueried object **Det**ection (**MQ-Det**), and introduces an efficient plug-in training architecture. The crux of the proposed MQ-Det is to fuse description-rich visual cues and the highly generalizable text representations, while only adding a small training cost on the basis of existing language queried fundamental detection models. We evaluate our models on a *finetuning-free* setting, where users can detect their customized objects through textual descriptions, visual exemplars, or both without any finetuning. With only one epoch of modulating on the Objects365 [36] dataset, accounting for a mere 3% of the total pre-training time for GLIP, our approach impressively improves the finetuning-free performance by +7.8% on the LVIS benchmark through providing the model with 5 visual exemplars along with textual category descriptions.

To be specific, as shown in Fig. 1, we first interleave a **Gated Class-scalable Perceiver** (**GCP**) that bridges class-wise visual cues and textual cues in each high-level stage of the text encoder of the detector. This module consists of class-wise cross attention layers to augment textual semantic embeddings with visual details, and a conditional gating layer that varies based on the quality of visual cues from the vision queries. The output of this module is incorporated to each high-level stage of the text encoder in a residual-like manner, which conducts textual perturbation in initial general embedding space [1]. Additionally, this simple design has no class-specific parameters and can be extended to classes of various granularity. Second, we design a **vision conditioned masked language prediction strategy** to ensure sufficient visual intervention in the modulating stage. We observed a learning inertia problem during modulating, namely, when incorporating visual cues in a gated residual-like manner, the learning process tends to be stuck around the initial optimum point of the frozen pre-training detector and cannot introduce much visual knowledge. Thus, we randomly mask the text tokens and let corresponding vision queries independently make object prediction. Note that we freeze the initial detection foundation model and only train the GCP modules in the modulating stage, which is quite efficient.

**Contributions**. In summary, the contributions of this paper are as follows: **(i)** As far as we know, we are the pioneer to introduce the multi-modal query that has the characteristics of both breadth and rich granularity, paving a new path to the object detection in the wild. **(ii)** We propose a plug-and-play GCP module that dynamically fuses informative visual cues and highly generalizable text cues from the multi-modal query, and employ a vision conditioned masked language prediction strategy to permit sufficient multi-modal fusion upon frozen detection models. **(iii)** Our proposed MQ-Det demonstrates powerful transferability on the finetuning-free and few-shot scenarios, while requiring much less training time than previous SoTA fundamental detectors. Specifically, MQ-Det outperforms GLIP by +7.8% AP in the finetuning-free detection on the challenging LVIS [13] and averagely +6.3% AP on 13 downstream few-shot detection tasks [23], modulated with merely 3% of the training time required by GLIP.

## 2 Methodology

This section describes MQ-Det: an efficient modulation architecture with elaborately designed multi-modal fusion strategy to let vision-language (VL) detection foundation models accept text interleaved with images as input queries, thus leveraging the generalization capability of semantic-rich language while enhancing detection discrimination with visual details. We first briefly review the language-queried baseline GLIP [25] in Sec. 2.1, then introduce the plug-and-play architecture that can support

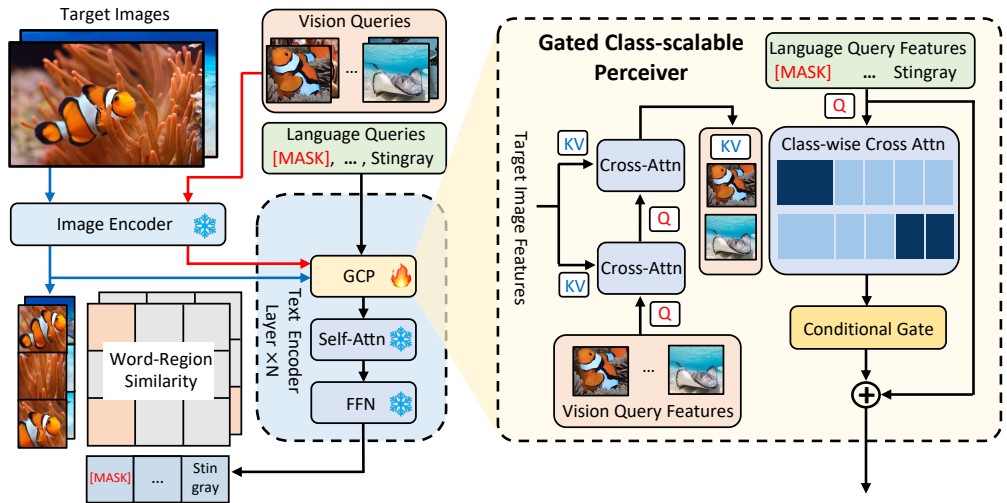

Figure 1: **Method overview**. The plug-and-play **G**ated **C**lass-scalable **P**erceivers (GCPs) are inserted to text encoder layers, trained with our masked language prediction strategy. Class-wise attention mask is applied to make the text query of each category only attends to corresponding vision queries.

class-scalable vision queries as input in Sec. 2.2, and finally shed light on the multi-modal fusion strategy in Sec. 2.3.

## 2.1 Preliminaries

**Language-queried detection model**. Following CLIP [32], the paralleled formulation has been a mainstream architecture of existing VL detection foundation models [48, 29, 42, 44, 28, 27]. Take our baseline GLIP [48] as an example, as shown in the left part of Fig. 1, it consists of paralleled text encoder $\Phi_T(\cdot)$ and image encoder $\Phi_I(\cdot)$ for feature extraction, and reformulates detection as a grounding task, by grounding/aligning each visual region to class content in a text prompt. Since the detection models like GLIP recognize objects via class content only in the language queries, we denote them as language-queried detection models.

Specifically, the model takes image-text pairs $\{(\mathcal{I}, \mathcal{T})\}$ as inputs, where the text prompt $\mathcal{T} = $ "$t_1, \ldots, t_{|C|}$" contains $|C|$ categories to be detected, *e.g.*, "person, bicycle, ... , toothbrush". Then, the model extracts image and text features $(I, T)$ via paralleled encoders. Every linear classification layer in traditional vision models are replaced by a vision-language matching dot-product layer, calculating the region-word similarity logits $S_{cls}$ in the detection head $H(\cdot)$. Formally,

$$I = \Phi_I(\mathcal{I}), \quad T = \Phi_T(\mathcal{T}), \quad R = H(I), \quad S_{cls} = R \cdot T^{\mathsf{T}}, \tag{1}$$

where $I$ is the extracted image features, $R \in \mathbb{R}^{N \times d}$ denotes the object/region/box features of the input image, and $T \in \mathbb{R}^{|C| \times d}$ is the contextual word/token features from the text encoder in which each token represents one category. Additionally, a box regressor is applied to $R$ for localization that is similar to traditional detectors. The model is finally trained with a phrase grounding loss calculated via the logits $S_{cls}$ and a traditional localization loss. We refer to GLIP [25] for more details.

The language-queried model GLIP is trained on massive image-text pairs, and thus scales up visual concepts beyond traditional detection vocabularies, and finally achieves strong open-set capacity. However, the inference process via a simple language-queried prompt limits its performance especially for non-descriptive, ambiguous or fine-grained class names.

## 2.2 Architecture design

Considering the limited description granularity of text stated in Sec. 1, we propose to integrate vision queries into the pre-trained language-queried detection models to enrich semantic cues. The architecture of such multi-modal queried detection models is expected to follow three principles: (1) **Class-scalable**: the introduced vision queries are open to arbitrary categories rather than fitting a

closed set of categories. (2) **Semantic-complementary**: the vision queries are sufficiently interleaved with coarse text queries to provide fine-grained visual details that support to distinguish various granularity of categories. (3) **Generalization-retainable**: the introduction of visual queries into foundation models accumulates rich details while still retaining the generalization capability.

We compare three possible architecture designs for enriching queries as illustrated in Fig. 2, and show the superiority of our approach on the above-mentioned principles. First, soft prompt based designs [53, 34, 4, 9] (e.g., CoOp [53], Fig. 2 (a)) can adapt text queries to describe diverse characteristics of categories beyond intensively-tuned manual prompts. However, most previous works only finetune task-specific prompts, which limits their generalization. Second, existing two-branch few-shot detection architectures [16, 21, 15, 40] (e.g., FCT [16], Fig. 2 (b)) support vision queries and detect novel objects using very few training exemplars. But their designs are incompatible with the language-queried pre-training foundation models due to deficiency of language inputs and model complexity (*e.g.*, early and deep fusion in [16]). We propose a simple yet effective plug-in architecture in Fig. 2 (c), which interleaves class-wise visual cues with textual cues, to let highly semantic language-queried models "perceive" visual details.

Inspired by [1], we augment the semantics of each category query by conditioning the text on corresponding visual representations produced by our proposed **Gated Class-scalable Perceiver (GCP)** illustrated in Fig. 1. Specifically, we insert GCP modules between the frozen pre-trained text encoder blocks in a residual manner and only train these modules from scratch. In a GCP module, each category token from the text encoder block is independently cross-attended to corresponding vision queries to acquire rich details (Principles 1,2). Then, the interleaved category tokens are gated by the GCP module so that the frozen text encoder is kept intact at initialization for improved stability and generalization performance (Principle 3).

Formally, given an image-text pair $(\mathcal{I}, \mathcal{T})$ with vision queries $\mathcal{V} = \{\mathbb{v}_i | \mathbb{v}_i = \{v_i^{(j)}\}_{j=1}^{k}\}_{i=1}^{|C|}$ extracted from $k$ exemplars of each category, the GCP module augments each text query features in $T = \Phi_T(\mathcal{T}) = \{t_i\}_{i=1}^{|C|}$ in a residual-like way, namely,

$$\bar{\mathbf{v}}_i = \text{X-MHA}(\mathbf{v}_i, I), \quad \hat{v}_i = \text{X-MHA}(t_i, \bar{\mathbf{v}}_i), \quad \hat{t}_i = t_i + \sigma(gate(\hat{v}_i)) \cdot \hat{v}_i, \quad i = 1, 2, \ldots, |C|, \quad (2)$$

where $\text{X-MHA}(\cdot, \cdot)$ denotes a multi-head cross-attention layer with the former input as queries and the later as keys and values. $\sigma = tanh(\cdot)$ is a normalization function. Concretely, the vision query features $\mathbf{v}_i = \Phi_I(\mathbb{v}_i)$ are first augmented via the target image feature $I = \Phi_I(\mathcal{I})$ to gather content-aware information [52]. The augmented vision queries $\bar{\mathbf{v}}_i$ are correlated with corresponding text token $t_i$ to enrich it from multiple views. The conditional gating value $\sigma(gate(\hat{v}_i))$ is dynamically adjusted according to the quality of visual cues from the exemplars of each category, which is evaluated by a three-layer perceptron (MLP) that reduces the feature dimension gradually to a layer-specific learnable scalar. The conditional gating value multiplies the enriched feature $\hat{v}_i$ before adding it to the initial text token $t_i$ from the residual connection. Since the gating value is initialized to 0, the module output matches that of the pre-trained text encoder at initial training, improving training stability and final performance. It is worth noting that this independent class-wise cross-attention scheme has no class-specific parameters and can be scaled to any classes of various description granularity, as verified in Sec. 3.2. In addition, it importantly allows the model to seamlessly generalize to any number of vision queries in inference, regardless of the number used during training.

### 2.3 Modulated pre-training

Equipped with the plug-in architecture design described in Sec. 2.2, we propose an efficient pre-training strategy that permits sufficient multi-modal fusion and rapid learning upon frozen detection foundation models using only a small subset of pre-training data. We denote this process as modulated pre-training in comparison to the massive pre-training of foundation models.

**Vision query extraction**. We provide vision queries for $|C|$ categories from a bank $D$ built from pre-training data:

$$D = \{\mathbf{v}_i | |\mathbf{v}_i| = K\}_{i=1}^{|C|}, \quad v_i^{(j)} = RoIPool(\Phi(\mathcal{I}), \gamma b), \quad j = 1, 2, \ldots, K, \quad (3)$$

where each $\mathbf{v}_i \in \mathbb{R}^{K \times d}$ contains $K$ vision queries modeled by the image encoder $\Phi_I$. Specifically, given a query instance of the $i$-th category with box $b \in \mathbb{R}^4$ in an image $\mathcal{I}$, an RoI pooler [33] is employed to extract the corresponding region feature $v_i^{(j)} \in \mathbb{R}^d$. The parameter $\gamma = 1.5^2$ enlarges the

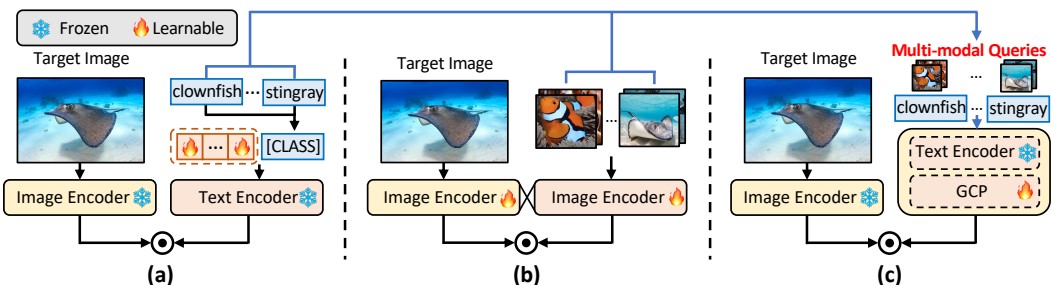

Figure 2: Architecture designs for query enriching, with only classification branches presented for clarity. (a) Prompt-based designs [53, 52] only tune upon language instruction while (b) two-branch few-shot detection architectures [16, 21] employ heavily coupled parallel encoders and merely take vision inputs. (c) MQ-Det elegantly combines both language and vision queries in an efficient way.

region area for contextual clues. During modulated pre-training, we randomly select $k \ll K$ vision queries from $D$ for each category at each forward process, simulating the downstream scenarios where only low-shot vision queries are provided. In practical implementation, we set $K = 5000$ and $k = 5$. After modulating, our model can generalize to arbitrary categories and vision queries during downstream inference.

**Training upon frozen pre-trained language-queried detectors**. Given the inherently general semantic representations in the pre-trained detector, we argue that the textual features only require minor modulation from the visual details rather than significant alteration. This viewpoint has been previously discussed in [52, 53, 1], and we have observed in Tab. 3 (c) that the full-model training on a limited number of categories faces the risk of catastrophic forgetting. Consequently, for efficiency purpose, we freeze the entire pre-trained detector and only train the newly-added gated class-scalable perceiver modules, significantly accelerating the training process.

**Vision conditioned masked language prediction**. Since the fundamental VL models highly rely on text description, the learning process of the residual-like gated architecture will rapidly converge around the initial optimum point with frozen text features, causing failure of vision queries. That is, models still learn how to align region with text features not visual cues. We represent such an issue as a learning inertia problem. As shown in the first row of Tab. 3 (a), the AP merely improves +0.9% over GLIP-T [25] on LVIS. To address this issue, we propose a simple yet effective masking strategy:

$$\mathcal{T} = \{t_1, t_2, \ldots, [\text{MASK}], \ldots, t_{|C|}\}. \tag{4}$$

Namely, given an image-text pair $(\mathcal{I}, \mathcal{T})$, a ground-truth text token in $\mathcal{T}$ corresponding to instances occurring in $\mathcal{I}$ is randomly masked by a [MASK] token with probability 40%. The [MASK] token is forced to extract visual cues from vision queries in the GCP modules to provide accurate predictions, thus bridging the model's dependence upon vision queries. As shown in Tab. 1, our language masking strategy ensures sufficient visual intervention in the modulated pre-training stage and significantly boosts the performance, *e.g.*, +4.4% AP over GLIP-T on finetuning-free LVIS.

## 3 Experiments

### 3.1 Setup

#### 3.1.1 Datasets and benchmarks

**Objects365 dataset** [36] is a large-scale, high-quality dataset for object detection. We use this dataset to conduct the modulated pre-training of our MQ-Det models. It contains 0.66 million images of 365 object categories, with 30 million bounding boxes, which are more diverse and fine-grained than those in other popular datasets such as COCO [26] and Pascal VOC [10]. Objects365 is a widely-used pre-training dataset in previous foundamental detection models [25, 48, 27, 50, 44, 42].

**LVIS benchmark** [13] is a challenging dataset for long-tail objects. It contains 1,203 categories for evaluation, with many rare categories that scarcely exist in the pre-training datasets. Therefore, we

Table 1: Finetuning-free detection on the LVIS benchmark. * denotes supervised approaches. The training time is tested on one V100 GPU. We present the number of vision queries during evaluation.

| Model | Backbone | Pre-Train Data | Data Size | Training Time (V100 days) | #Vision Query | MiniVal (%) | | | | Val v1.0 (%) | | | |
|---|---|---|---|---|---|---|---|---|---|---|---|---|---|
| | | | | | | AP | $AP_r$ | $AP_c$ | $AP_f$ | AP | $AP_r$ | $AP_c$ | $AP_f$ |
| MDETR [20]* | RN101 | GoldG,RefC | 0.9M | 400 | 0 | 24.2 | 20.9 | 24.9 | 24.3 | 22.5 | 7.4 | 22.7 | 25.0 |
| Mask R-CNN [17]* | RN101 | - | - | - | 0 | 33.3 | 26.3 | 34.0 | 33.9 | - | - | - | - |
| Supervised-RFS [13]* | RN50 | - | - | - | 0 | - | - | - | - | 25.4 | 12.3 | 24.3 | 32.4 |
| GLIP-T (B) [25] | Swin-T | O365 | 0.66M | 300 | 0 | 17.8 | 13.5 | 12.8 | 22.2 | 11.3 | 4.2 | 7.6 | 18.6 |
| GLIP-T [25] | Swin-T | O365,GoldG,CC4M | 5.5M | 480 | 0 | 26.0 | 20.8 | 21.4 | 31.0 | 17.2 | 10.1 | 12.5 | 25.5 |
| GLIPv2-T [48] | Swin-T | O365,GoldG,CC4M | 5.5M | - | 0 | 29.0 | - | - | - | - | - | - | - |
| GroundingDINO-T [27] | Swin-T | O365,GoldG,Cap4M | 5.5M | - | 0 | 25.7 | 15.2 | 21.9 | 30.9 | - | - | - | - |
| GLIP-L [25] | Swin-L | FourODs,GoldG,Cap24M | 27.5M | 600 | 0 | 37.3 | 28.2 | 34.3 | 41.5 | 26.9 | 17.1 | 23.3 | 35.4 |
| GroundingDINO-L [27] | Swin-L | O365,OI,GoldG,Cap4M,COCO,RefC | 15.8M | - | 0 | 33.9 | 22.2 | 30.7 | 38.8 | - | - | - | - |
| MQ-GLIP-T-Img | Swin-T | O365† | 0.66M | 10 | 5 | 17.6 | 12.0 | 14.5 | 21.2 | 12.4 | 8.9 | 9.2 | 18.3 |
| MQ-GLIP-T-Txt | Swin-T | O365† | 0.66M | 10 | 0 | 26.0 | 20.8 | 21.4 | 31.0 | 17.2 | 10.1 | 12.5 | 25.5 |
| MQ-GroundingDINO-T | Swin-T | O365† | 0.66M | 10 | 5 | 30.2 | 21.7 | 26.2 | 35.2 | 22.1 | 12.9 | 17.4 | 31.4 |
| MQ-GLIP-T | Swin-T | O365† | 0.66M | 10 | 5 | 30.4 | 21.0 | 27.5 | 34.6 | 22.6 | 15.4 | 18.4 | 30.4 |
| MQ-GLIP-L | Swin-L | O365† | 0.66M | 22 | 5 | 43.4 | 34.5 | 41.2 | 46.9 | 34.7 | 26.9 | 32.0 | 41.3 |

use LVIS in the downstream task to evaluate the finetuning-free transferability. We report on MiniVal containing 5,000 images introduced in MDETR [20] as well as the full validation set v1.0.

**ODinW benchmark** [23] (Object Detection in the Wild) is a more challenging benchmark for evaluating model performance under real-world scenarios. For example, Aquarium requires locating fine-grained fish species, and Pothole concerns detecting holes on the road. It collects more than 35 datasets for evaluation. There are two commonly used versions of the ODinW benchmark, *i.e.*, ODinW-13 and ODinW-35, where ODinW-13 is a subset of ODinW-35 and contains much less noisy data. The results on both benchmarks are reported for comprehensive comparison. We demonstrate that the fine-grained visual details in MQ-Det facilitate transfer to such diverse and challenging tasks.

### 3.1.2 Implementation details

We conduct extensive experiments on three settings: an open-set setting on finetuning-free LVIS [13] and ODinW [23] benchmarks, a few-shot setting and a close-set full-shot setting on ODinW. To demonstrate the plug-and-play versatility of our approach, we apply MQ-Det on two typical SoTA language-queried object detectors, GLIP [25] and GroundingDINO [27], and obtain our multi-modal queried models **MQ-GLIP** and **MQ-GroundingDINO**, respectively. We incorporate our GCP modules into the last 6 layers of the frozen text encoder. We conduct modulated pre-training of our models on the Objects365 dataset [36] for only one epoch using 8 NVIDIA V100 GPUs. The finetuning process under few/full-shot settings completely follows the baseline method GLIP, where vision queries are extracted from the few/full-shot training set. We also evaluate our method in a *finetuning-free* setting, namely, users can detect their customized objects through textual descriptions, visual exemplars, or both without any fine-tuning. During finetuning-free evaluation, we extract 5 instances as vision queries for each category from the downstream training set without any finetuning.

### 3.2 MQ-Det helps low-shot object detection

### 3.2.1 Multi-modal queried detection without finetuning

We evaluate the model's ability to recognize rare and diverse objects on both LVIS and ODinW in a *finetuning-free* setting, where users can detect their customized objects through textual descriptions, visual exemplars, or both without any fine-tuning. Each category is provided with both language and vision queries. Tab. 1 shows the results on LVIS. Overall, MQ-Det demonstrates **strong finetuning-free transferability** with impressive **efficiency on both data usage and training time**. MQ-GLIP-L surpasses the current SoTA by a large margin with simply 5 visual exemplars provided, reaching 34.7% AP on Val v1.0 (+7.8% over GLIP-L), which verifies the superiority of multi-modal queries over single-modal queries. Meanwhile, MQ-Det demonstrates good training efficiency, *e.g.*, MQ-GLIP-T only requires additional 2% training time and 12% data usage when modulated upon GLIP-T.

Additionally, we find three contributing factors: **(i)** Strong generalization comes from combination of textual breadth and visual granularity. In Tab. 1, MQ-GLIP-T-Img replaces all text queries with [MASK] tokens and solely utilizes vision queries to predict objects, achieving 17.6% AP.

---

†Modulating upon pretrained models indirectly utilizes their pre-training data.

Table 2: Results on the ODinW benchmark. The few-shot performance is evaluated by 3-shot [27, 25].

| Model | Language Query | Vision Query | Backbone | Pre-train Data | Data Size | ODinW-35 $AP_{avg}$ | ODinW-13 $AP_{avg}$ |
|---|---|---|---|---|---|---|---|
| *Finetuning-free Setting* | | | | | | | |
| MDETR [20] | ✓ | ✗ | ENB5 [38] | GoldG,RefC | 0.9M | 10.7 | 25.1 |
| OWL-ViT [30] | ✓ | ✓ | ViT L/14(CLIP) | O365, VG | 0.8M | 18.8 | 40.9 |
| GLIP-T [25] | ✓ | ✗ | Swin-T | O365,GoldG,Cap4M | 5.5M | 18.7 | 41.9 |
| GLIP-L [25] | ✓ | ✗ | Swin-L | FourODs,GoldG,Cap24M | 27.5M | 22.6 | 51.0 |
| OmDet [50] | ✓ | ✗ | ConvNeXt-B | COCO,O365,LVIS,PhraseCut | 1.8M | 16.0 | 43.6 |
| GLIPv2-T [48] | ✓ | ✗ | Swin-T | O365,GoldG,Cap4M | 5.5M | 22.3 | 50.7 |
| DetCLIP [42] | ✓ | ✗ | Swin-T | O365,GoldG,YFCC1M | 2.4M | - | 43.3 |
| GroundingDINO-T [27] | ✓ | ✗ | Swin-T | O365,GoldG,Cap4M | 5.5M | 21.7 | 49.8 |
| MQ-GroundingDINO-T | ✓ | ✓ | Swin-T | O365† | 0.66M | 22.5 | 50.9 |
| MQ-GLIP-T | ✓ | ✓ | Swin-T | O365† | 0.66M | 20.8 | 45.6 |
| MQ-GLIP-L | ✓ | ✓ | Swin-L | O365† | 0.66M | **23.9** | **54.1** |
| *Few-Shot Setting* | | | | | | | |
| DyHead-T [6] | ✗ | ✗ | Swin-T | O365 | 0.66M | 37.5 | 43.1 |
| GLIP-T [25] | ✓ | ✗ | Swin-T | O365,GoldG,Cap4M | 5.5M | 38.9 | 50.7 |
| DINO-Swin-T [47] | ✗ | ✗ | Swin-T | O365 | 0.66M | 41.2 | 49.0 |
| OmDet [50] | ✓ | ✗ | ConvNeXt-B | COCO,O365,LVIS,PhraseCut | 1.8M | 42.4 | 48.5 |
| MQ-GLIP-T | ✓ | ✓ | Swin-T | O365† | 0.66M | **43.0** | **57.0** |
| *Full-Shot Setting* | | | | | | | |
| GLIP-T [25] | ✓ | ✗ | Swin-T | O365,GoldG,Cap4M | 5.5M | 62.6 | 61.9 |
| DyHead-T [6] | ✗ | ✗ | Swin-T | O365 | 0.66M | 63.2 | 58.7 |
| DINO-Swin-T [47] | ✗ | ✗ | Swin-T | O365 | 0.66M | 66.7 | - |
| OmDet [50] | ✓ | ✗ | ConvNeXt-B | COCO,O365,LVIS,PhraseCut | 1.8M | 67.1 | 65.3 |
| DINO-Swin-L [47] | ✗ | ✗ | Swin-L | O365 | 0.66M | 68.8 | 67.3 |
| MQ-GLIP-T | ✓ | ✓ | Swin-T | O365† | 0.66M | 64.8 | 62.5 |

Table 3: Ablation results. We evaluate finetuning-free performance on LVIS and ODinW. Both average and median values on 13 datasets of ODinW are reported. Each row in this ablation study table should be compared to the baseline MQ-GLIP-T reported at the top of the table.

| | Ablated Setting | Ablated Details | MQ-GLIP-T Original Value | → Changed Value | LVIS MiniVal (%) AP | $AP_r$ | $AP_c$ | $AP_f$ | ODinW-13 (%) $AP_{avg}$ | $AP_{mid}$ |
|---|---|---|---|---|---|---|---|---|---|---|
| | Original GLIP-T model | | | | 26.0 | 20.8 | 21.4 | 31.0 | 41.9 | 39.3 |
| | **MQ-GLIP-T model** | | | | **30.4** | **21.0** | **27.5** | **34.6** | **45.6** | **48.8** |
| **(a)** | Mask Strategy | Mask Rate | 40% | 0% | 26.9 | 20.1 | 23.4 | 31.2 | 42.8 | 46.8 |
| | | | 40% | 80% | 28.9 | 20.4 | 24.2 | 34.6 | 45.0 | 48.3 |
| **(b)** | Gate | Enable | ✓ | ✗ | 27.1 | 19.4 | 24.4 | 31.1 | 44.1 | 47.1 |
| | | Architecture | MLP | Scaler | 29.3 | 18.3 | 26.3 | 34.0 | 43.4 | 46.0 |
| | | | MLP | Linear | 30.0 | 19.7 | 26.9 | 34.5 | 44.4 | 47.3 |
| | | Input | $\hat{v}$ | $cat(\hat{v}, t)$ | 29.2 | 17.8 | 26.0 | 34.2 | 44.8 | 48.2 |
| **(c)** | Freezing | Full Model | ✓ | ✗ | 24.4 | 28.6 | 21.1 | 17.2 | 38.6 | 45.4 |
| | | Detector Text Encoder | ✓ | ✗ | 26.6 | 17.3 | 23.0 | 31.5 | 42.7 | 46.6 |

This confirms that our low-cost modulated pre-training strategy introduces sufficient visual cues. Augmenting generalizable language with visual details, MQ-GLIP-T outperforms the text-only model MQ-GLIP-T-Txt by a large margin. **(ii)** Efficiency on data usage and training time benefits from frozen pre-trained detection models. Our approach preserves the generalizable knowledge in the frozen pre-trained models and thus only needs little additional cost for further modulation. This is also verified in Tab. 3 (c). **(iii)** Our simple plug-and-play architecture design enables wide-range applications, since most language-queried detectors share the similar parallel architecture design as GLIP and GroundingDINO, *e.g.*, MDETR [20], OWL-ViT [30], and DetCLIP [42] in Tab. 2.

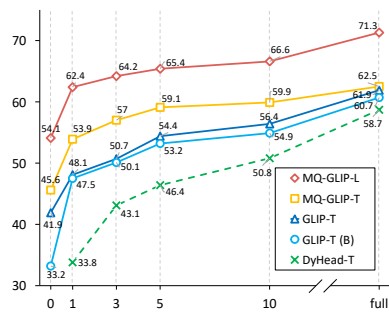

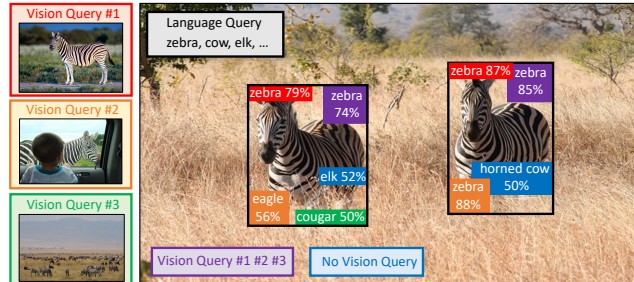

Figure 3: Average precision (%) on ODinW-13, from finetuning-free to full-shot data. MQ-Det clearly improves GLIP on data efficiency.

Figure 4: Finetuning-free detection with different vision queries. The language queries of 1,203 LVIS categories are always provided. Similar boxes are located while vision queries with better quality provide more accurate predictions.

#### 3.2.2 MQ-Det as a strong few-shot learner

The experiments on ODinW with respect to finetuning-free, few-shot and full-shot settings are listed in Tab. 2, where MQ-Det shows robust performance on this challenging benchmark. We observe that most large-scale fundamental detection models only support language as category queries. With additional vision queries, MQ-GLIP-T averagely improves GLIP-T by +3.7% AP and +6.3% AP in the finetuning-free and few-shot settings respectively on ODinW-13, thus exhibiting the superiority of the rich description granularity of vision queries in our approach. Meanwhile, MQ-GLIP-T sets a new record on the ODinW-35 few-shot setting with 43.0% AP. Fig. 3 demonstrates few-shot performance with respect to different amounts of training data. We also notice that MQ-GLIP-T makes limited gains over GLIP-T with sufficient training data in the full-shot setting (*e.g.*, +0.6% AP on ODinW-13). The reason is that the learning procedure, with sufficient training data, degenerates to a traditional close-set detection problem, where the model just represents categories as index regardless of textual or visual descriptions.

### 3.3 Ablation studies

**Importance of the masked language prediction strategy.** We propose our masked language prediction strategy to address the learning inertia problem described in Sec. 2.3. Tab. 3 (a) verifies the effect of our pre-training strategy. We observe that our approach holds a great performance degradation (-3.5% AP) on LVIS with the mask rate of 0. MQ-GLIP-T with the mask rate of 0 only improves GLIP-T by +0.9% AP on LVIS. This indicates that our multi-modal queried MQ-GLIP may lose its effect and degenerate to language-queried GLIP without the masked language prediction strategy. Meanwhile, a too large mask rate (*e.g.*, 80%) may lead to overfitting on the vision queries and losing attention to semantic-rich language queries, thus impairing performance.

**The role of vision queries**. MQ-Det provides users a flexible way to construct customized vision queries. We use one vision query for each category of LVIS in MQ-GLIP-T, and investigate the effect of different query qualities as shown in Fig. 4. The vision queries are selected based on the following criteria: **(i)** Positive queries (in red) contain complete information about the target objects, which significantly improve detection. **(ii)** Hard positive queries (in brown) only provide partial object information and sometimes include additional noise, such as the example of the boy and car. Despite the additional noise, MQ-Det can filter it out and still help improve the detection accuracy to some extent, identifying at least one object correctly. **(iii)** Negative queries (in green) contain too much noise and do not benefit detection, while **(iv)** "no vision queries" (in blue) is used as a baseline. Finally, we also use **(v)** mixed queries (in purple) that combine the three types of vision queries. The results show the robustness of MQ-Det, namely, our approach can select relevant information from miscellaneous vision queries to boost detection.

**Architecture design**. We design a plug-and-play GCP module so that visual details can be smoothly interleaved with highly-semantic textual cues. Particularly, in GCP, the conditional gating mechanism on each interleaved category token in Eqn. (2) ensures smooth model initialization from the frozen model and stable modulating according to different quality of visual cues. As shown in the first two

rows of Tab. 3 (b), roughly adding visual cues to each text token, or equally scaling them using the same learnable gating scalar for all vision queries, can result in sub-optimal performance (-3.3% AP and -1.1% AP on LVIS, respectively). When using a simple gating design consisting of one linear layer over different vision queries, we obtain a slight performance drop compared to the non-linear MLP design, indicating content-aware gating mechanism is necessary and more parametric design helps analyze the quality of vision queries. In order to encourage the gate to balance the vision query $\hat{v}$ and the text query $t$ in Eqn. (2), we combine the two query types as a joint input (the last row of Tab. 3 (b)). However, we discover that this task rises the learning difficulty for a simple gate design.

**Freezing pre-trained detectors**. Tab. 3 (c) shows the necessity of modulating on the frozen models to introduce vision queries into the language-queried detectors. Either modulating the whole pre-trained detector or only modulating the text encoder with the limited detection data inevitably lose the generalization capability of the initial language-queried model, leading to a significant performance decrease (-6.0% AP and -3.8% AP on LVIS respectively). Moreover, training without freezing the detector requires almost twice the time consumption, not to mention the additional memory cost.

## 4 Related work

**Language-queried object detection.** Recently, vision-and-language approaches have gained popularity in visual recognition, where language is used to query visual objects and enable open-set ability. CLIP [32] and ALIGN [19] are pioneers in cross-modal image-level contrastive learning [5], where they train on millions of image-text pairs to perform open-vocabulary image classification. Inspired by these works, object detection has pursued cross-modal region-level contrastive learning in two main pipelines. One pipeline is open-vocabulary object detection (OVD) [46, 12, 54, 51, 30, 45], where the detection data is rigorously split into base classes for training and novel classes for evaluation. For instance, Zareian *et al.* [46] propose OVR-CNN that first pre-trains a visual encoder on image-caption pairs to learn rich vocabulary and then finetunes on detection data with only base classes. Detic [54] learns object localization and fine-grained vision-language matching simultaneously using max-size proposals to assign image-level labels. However, this pipeline utilizes limited detection data and only generalizes to specific datasets, *e.g.*, from COCO [26] base classes to COCO novel classes. In contrast, our approach follows the second pipeline of large-scale fundamental detection models [42, 25, 48, 20, 50], which aims to generalize to arbitrary categories with extensive region-level image-text data. MDETR [20] trains an end-to-end DETR [3] model via phrase-region matching on existing multi-modal datasets, providing a strong baseline for few-shot object detection. GLIP [25, 48] formulates object detection as a grounding problem and leverages additional grounding data for aligned semantics at phrase and region levels, achieving even stronger performance on fully-supervised detection benchmarks without finetuning. GroundingDINO [27] applies GLIP's grounded training strategy to a stronger detection backbone DINO [47], achieving state-of-the-art open-set detection performance. Since these models all use a text encoder like BERT [8] to model the language queries, our approach can seamlessly integrate vision queries with these language-queried models to enable rich visual granularity.

**Vision-queried methods** commonly exist in current two-branch based few-shot object detection pipelines [16, 11, 14, 15, 21]. These methods are based on a siamese network to process the target images and vision queries in parallel, and calculate the similarity between image regions (usually proposals) and few-shot examples for detection. Kang *et al.* [21] propose a feature reweighting module to aggregate the target image features via few-shot vision queries. Meta Faster R-CNN [15] performs feature alignment between the two inputs and focus on foreground regions using attention. FCT [16] proposes an early fusion transformer architecture to fuse target images and vision queries. However, the two-branch pipeline in few-shot detection lacks instruction of language, and is far from practical implementation, *e.g.*, no more than 10% AP in 3-shot COCO [26] for current SoTA methods [16, 15], not to mention more challenging LVIS [13] and ODinW [23] benchmarks. In addition, only a very few open-vocabulary object detection approaches support both language and vision queries. OV-DETR [45] proposes to utilize CLIP [32] embeddings from both image and text encoders as the object queries of the DETR [3] decoder. OWL-ViT [30] uses detection data to finetune upon CLIP. Thanks to the parallel structure of CLIP, OWL-ViT can conduct both region-text and region-image similarity. However, these two methods do not support language and vision queries as joint inputs, and show low performance with vision queries. Our MQ-Det considers the

complementary nature of vision and language queries, thus achieving both open-set generalization and fine-grained recognition.

## 5 Conclusion

MQ-Det is an efficient pre-training architecture and strategy design to let vision-language (VL) detection foundation models accept text interleaved with images as the input queries. After modulated pre-training, MQ-Det shows promising gains on the finetuning-free and few-shot settings on well-established benchmarks and 35 downstream tasks. Our results suggest that utilizing the complementary nature of language and vision is an important step towards open-world object detection.

## 6 Acknowledgements

This work was supported by Key-Area Research and Development Program of Guangdong Province (2021B0101400002), National Natural Science Foundation of China (No. 62036012, 62322212, 62072455, 61721004), and by Tencent Rhino-Bird Research Elite Program.

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

# Appendix

We provide an overview of the Appendix below.

**Transfer to downstream (Sec. A).** We elaborate on the additional details to transfer our MQ-Det to downstream tasks, including finetuning-free, few-shot, and full-shot settings. The introduction is organized as followed.

- Different ways to acquire the vision queries in Sec. A.1. This is an elaborative description of Sec. 3.1.2 in the main text.
- A customized tuning approach of MQ-Det in Sec. A.3, which achieves similar performance as full-model tuning while only fine-tuning very few parameters.

**Experiments (Sec. B).** We provide additional training and evaluation details, including:

- Detailed results on 35 downstream tasks in ODinW are provided in Tab. III and Tab. VII, described in Sec. B.1.
- An explicit evaluation on an open-vocabulary detection setting is conducted to further investigate the generalization of our multi-modal queries, as presented in Sec. B.2.
- Some visualization results of MQ-Det in Sec. B.1 and Fig. I.
- The training hyper-parameters are illustrated in Sec B.3 and Tab. VI.

**Further discussion (Sec. C).** We provide a comprehensive discussion on our work, including limitations and broader impacts in Sec C.

## A  Transfer to downstream

### A.1  Different ways to acquire vision queries

During finetuning-free evaluation, we extract 5 instances as vision queries for each category from the downstream training set without any finetuning. We also provide two alternative strategies and observe similar performance, *i.e.*, retrieval and test-time online update, where the former obtains vision queries from heterogeneous external data like ImageNet [7], and the latter dynamically stores high-confidence instances as vision queries during evaluation. These two additional approaches are proposed to simulate realistic scenarios:

- Retrieval (user-provided exemplars): a small number of exemplars are provided by the users without any fine-tuning. We retrieve 5 exemplars as vision queries for each category from ImageNet-21K [7] to simulate the user-provided exemplars. These samples are heterogeneous from the downstream test data, *e.g.*, domains. The results are provided in MQ-GLIP-T-Retrival of Tab. I.
- Online updating: the model dynamically stores high-confidence instances as vision queries during evaluation. No vision queries are provided at the initial stage of evaluation. The results are illustrated in MQ-GLIP-T-Online of Tab. I. We provide detailed description in Sec. A.2.

The results are illustrated in Tab. I. We select 3 downstream datasets from ODinW [23] to verify the effectiveness of each approach. Generally, all three approaches to acquire vision queries demonstrate similar performance. We observe that vision queries from online updating hold relatively lower quality, thus leading to slight performance drop, since no manual annotations are provided. Meanwhile, exemplars retrieved from ImageNet are object-centric and contain little noise (*e.g.*, other irrelevant objects), thus improving the performance.

### A.2  Test-time online update

Our test-time online update strategy is conducted via the following steps: 1) only utilize language queries to conduct detection at the initial stage of evaluation. 2) Store detected instances with high confidence as the vision queries of corresponding categories. 3) Use both language queries and stored

vision queries for evaluation and seek for more vision queries. Tab. I verifies the effectiveness of our approach.

### A.3 Partial tuning rivals full-model tuning

Since the GCP modules are interleaved into the frozen detector, we provide a customized fine-tuning strategy, partial tuning, namely, only tuning the newly added GCP modules and freezing all other parameters. The results in Tab. II indicate that our partial tuning strategy achieves comparable performance with traditional full-model tuning (*i.e.*, only -0.4 % AP on ODinW-13). Partial tuning accounts for much fewer learnable parameters, thus friendly to training time and memory costs. The experimental results in the main text are all based on the partial tuning.

Table I: Different ways to acquire vision query. We report the finetuning-free performance. All models should be compared with the MQ-GLIP-T model at the top of the table.

| Model | AerialDrone | Aquarium | Rabbits |
|---|---|---|---|
| GLIP-T | 12.5 | 18.4 | 70.2 |
| MQ-GLIP-T | 15.8 | 23.5 | 75.4 |
| MQ-GLIP-T-Online | 15.5 | 23.2 | 74.8 |
| MQ-GLIP-T-Retrieval | 16.0 | 23.6 | 75.1 |

Table II: Different fine-tuning strategies of MQ-Det under the 5-shot setting. The implementation used in our evaluation is highlighted in color.

| Strategy | ODinW-13 (%) | |
|---|---|---|
| | $AP_{avg}$ | $AP_{mid}$ |
| Partial tuning | 59.1 | 62.4 |
| Full-model tuning | 59.5 | 64.5 |

## B Experiments

### B.1 All results

We report the per-dataset performance under various settings in ODinW-13 and ODinW-35, shown in Tab. III and Tab. VII, respectively. We also provide some visualized results in Fig. I.

Table III: Per-dataset AP performance (%) on ODinW-13. We report results on 0, 1, 3, 5, 10-shot detection. MQ-GD-T denotes MQ-GroundingDINO-T. Specifically, the "zero-shot" here actuall stands for the finetuning-free setting with 5 vision queries.

| Dataset | MQ-GLIP-T | | | | | MQ-GLIP-L | | | | | MQ-GD-T |
|---|---|---|---|---|---|---|---|---|---|---|---|
| | 0 | 1 | 3 | 5 | 10 | 0 | 1 | 3 | 5 | 10 | 0 |
| PascalVOC | 59.8 | 52.9 | 58.9 | 59.3 | 59.6 | 64.7 | 64.7 | 67.4 | 68.1 | 68.8 | 57.5 |
| AerialDrone | 15.8 | 22.1 | 29.8 | 31.0 | 31.6 | 17.4 | 30.7 | 36.1 | 37.0 | 36.1 | 13.6 |
| Aquarium | 23.5 | 31.7 | 36.1 | 40.1 | 42.4 | 30.3 | 39.2 | 45.8 | 47.0 | 49.7 | 18.5 |
| Rabbits | 75.4 | 76.2 | 77.4 | 75.6 | 75.5 | 71.8 | 76.0 | 75.0 | 74.3 | 75.3 | 79.9 |
| EgoHands | 41.2 | 64.3 | 66.0 | 66.7 | 68.2 | 57.2 | 68.8 | 68.1 | 71.6 | 72.6 | 65.4 |
| Mushrooms | 61.0 | 89.7 | 89.0 | 91.8 | 89.0 | 63.9 | 87.4 | 91.6 | 90.7 | 92.2 | 68.2 |
| Packages | 68.5 | 71.9 | 72.8 | 73.7 | 74.4 | 53.0 | 70.6 | 71.2 | 72.0 | 73.5 | 64.1 |
| Raccoon | 41.6 | 61.2 | 64.8 | 65.5 | 61.9 | 58.1 | 70.9 | 73.3 | 72.0 | 76.7 | 49.2 |
| Shellfish | 26.6 | 27.9 | 34.2 | 41.9 | 40.0 | 63.0 | 61.1 | 60.1 | 62.8 | 60.7 | 29.2 |
| Vehicles | 57.2 | 60.6 | 59.5 | 65.7 | 65.6 | 63.2 | 68.3 | 70.2 | 71.2 | 72.4 | 56.7 |
| Pistols | 59.6 | 56.5 | 60.3 | 61.4 | 61.7 | 74.4 | 73.6 | 72.7 | 74.3 | 74.8 | 69.2 |
| Pothole | 14.7 | 26.7 | 28.0 | 33.7 | 36.4 | 27.0 | 30.9 | 30.8 | 36.9 | 38.7 | 25.2 |
| Thermal | 48.0 | 59.4 | 64.2 | 62.4 | 72.7 | 58.7 | 68.5 | 72.2 | 72.5 | 74.5 | 64.9 |
| Average | 45.6 | 53.9 | 57.0 | 59.1 | 59.9 | 54.1 | 62.4 | 64.2 | 65.4 | 66.6 | 50.9 |

### B.2 Explicit evaluation on an open-vocabulary detection setting

To further investigate the transferability of MQ-Det, we evaluation our models on a clear separation of base and novel classes, which is similar to previous open-vocabulary object detection [54, 41]. We first construct a novel category set from 1,203 LVIS categories. Specifically, we remove the LVIS categories that exist in the 365 classes of Objects365 and finally obtain 986 novel categories that

did not appear during our modulated pre-training. The remaining 217 categories are represented as base categories. Then, we conduct finetuning-free inference with 5 vision queries on the separated categories to verify the generalization of multi-modal query learning. Tab. IV shows the results. The results indicate that multi-modal queries generalize well to novel classes that do not exist in the modulated pre-training. Specifically, +4.1%, +5.7%, and +6.3% AP on novel classes of MQ-GroundingDINO-T, MQ-GLIP-T, and MQ-GLIP-L over their baselines, respectively.

Table IV: Finetuning-free detection with explicit open-vocabulary category separation on LVIS.

| Model | $AP_{novel}$ | $AP_{base}$ | $AP_{all}$ |
|---|---|---|---|
| GroundingDINO-T | 22.1 | 36.7 | 25.6 |
| GLIP-T | 20.8 | 42.0 | 26.0 |
| GLIP-L | 35.4 | 45.5 | 37.9 |
| MQ-GroundingDINO-T | 26.2 | 43.0 | 30.2 |
| MQ-GLIP-T | 26.5 | 42.8 | 30.4 |
| MQ-GLIP-L | 41.7 | 51.3 | 44.0 |

It is worth noting that the separation of base and novel classes differs from previous works on open-vocabulary detection (OVD) [46]. The reason is that the testing categories of previous separation are partially included in our pre-training dataset Objects365 [36]. Therefore, we represent the classes in LVIS that do not exist in our modulated pre-training dataset Objects365 as novel classes. The frequency distribution of the separated LVIS dataset is shown in Tab. V:

Table V: Frequency distribution of the separated LVIS for open-vocabulary evaluation.

| Class | #Rare | #Common | #Frequent |
|---|---|---|---|
| Novel | 326 | 404 | 256 |
| Base | 11 | 57 | 149 |
| All | 337 | 461 | 405 |

### B.3 Training hyper-parameters

We report the hyper-parameter settings of the modulated pre-training of MQ-Det in Tab. VI. Other settings are the same with corresponding language-queried detectors.

Table VI: Hyper-parameters of modulated pre-training.

| Item | Value | Item | Value |
|---|---|---|---|
| optimizer | AdamW | max vision query num ($K$) | 5000 |
| lr of GCP | 1e-5 | vision query num ($k$) | 5 |
| lr of gate | 5e-3 | mask rate | 40% |
| weight decay | 1e-4 | layer with GCP | 6~12 |

## C   Further discussion

**Limitations**. First, multi-modal queries make limited contribution with sufficient training data for each category. This may because the foundation models learn enough accurate classification boundaries, thus reducing the effectiveness of language and vision queries. Second, the applications of MQ-Det on other dense prediction tasks such as segmentation remain unexplored.

**Broader impacts**. MQ-Det shows strong downstream transfer ability with highly flexible category vocabularies. This allows inexperienced users to easily use MQ-Det models (*e.g.*, MQ-GLIP) for their own needs by simply providing some visual examples and corresponding text descriptions. However, this also raises concerns about how our MQ-Det models with a large vocabulary could be used inappropriately in the community, such as for large-scale illegal video surveillance. The

open-set detection capabilities could be manipulated through specialized visual or textual cues to facilitate targeted detections instead of generic ones. This manipulation could introduce biases in the detector and result in unfair predictions.

Table VII: Per-dataset AP performance (%) on ODinW-35. We report results on 0, 3-shot detection. MQ-GD-T denotes MQ-GroundingDINO-T. Specifically, the "zero-shot" here actuall stands for the finetuning-free setting with 5 vision queries.

| Dataset | MQ-GLIP-T | | MQ-GLIP-L | MQ-GD-T |
| | 0 | 3 | 0 | 0 |
| --- | --- | --- | --- | --- |
| AerialMaritimeDrone_large | 15.8 | 29.8 | 17.4 | 13.6 |
| AerialMaritimeDrone_tiled | 18.3 | 27.1 | 20.8 | 21.9 |
| AmericanSignLanguageLetters | 1.8 | 19.9 | 3.0 | 0.1 |
| Aquarium | 23.5 | 36.1 | 30.3 | 18.5 |
| BCCD_BCCD | 4.3 | 57.3 | 12.4 | 12.3 |
| ChessPiece | 10.7 | 64.8 | 2.8 | 14.9 |
| CottontailRabbits | 75.4 | 77.4 | 71.8 | 79.9 |
| DroneControl_Drone_Control | 6.4 | 40.9 | 9.3 | 1.2 |
| EgoHands_generic | 41.2 | 66.0 | 57.2 | 65.4 |
| EgoHands_specific | 3.6 | 32.1 | 6.7 | 0.1 |
| HardHatWorkers | 5.4 | 37.4 | 5.5 | 4.8 |
| MaskWearing | 0.3 | 46.3 | 1.0 | 0.0 |
| MountainDewCommercial | 49.2 | 46.4 | 17.7 | 39.9 |
| NorthAmericaMushrooms | 61.0 | 89.0 | 63.9 | 68.2 |
| OxfordPets_by-breed | 0.4 | 31.5 | 0.5 | 0.4 |
| OxfordPets_by-species | 1.6 | 68.1 | 0.4 | 0.5 |
| PKLot_640 | 0.9 | 30.3 | 2.6 | 0.0 |
| Packages | 68.5 | 72.8 | 53.0 | 64.1 |
| Raccoon_Raccoon | 41.6 | 64.8 | 58.1 | 49.2 |
| ShellfishOpenImages | 26.6 | 34.2 | 63.0 | 29.2 |
| ThermalCheetah | 2.3 | 41.1 | 11.1 | 7.1 |
| UnoCards | 0.2 | 35.4 | 0.8 | 0.0 |
| VehiclesOpenImages | 57.2 | 59.5 | 63.2 | 56.7 |
| WildFireSmoke | 13.2 | 22.5 | 20.5 | 14.6 |
| boggleBoards | 0.1 | 76.5 | 0.1 | 0.0 |
| brackishUnderwater | 4.5 | 31.3 | 5.3 | 3.3 |
| dice_mediumColor | 0.4 | 14.6 | 0.6 | 0.1 |
| openPoetryVision | 0.0 | 3.1 | 0.0 | 0.1 |
| pistols | 59.6 | 60.3 | 74.4 | 69.2 |
| plantdoc | 1.7 | 12.5 | 1.5 | 0.2 |
| pothole | 14.7 | 28.0 | 27.0 | 25.2 |
| selfdrivingCar | 8.2 | 17.5 | 9.0 | 6.9 |
| thermalDogsAndPeople | 48.0 | 64.2 | 58.7 | 64.9 |
| websiteScreenshots | 1.0 | 6.6 | 1.7 | 1.0 |
| PascalVOC | 59.8 | 58.9 | 64.7 | 57.5 |
| Average | 20.8 | 43.0 | 23.9 | 22.5 |

Figure I: Visualized results of MQ-GLIP-T on the LVIS benchmark.

Prediction    Annotation    Prediction    Annotation

