# OpenReview forum: "Multi-modal Queried Object Detection in the Wild"
_NeurIPS.cc/2023/Conference — NeurIPS 2023 poster_

### Official Review · Reviewer_wDpN · 2023-07-06

**Soundness:** 3 good
**Presentation:** 3 good
**Contribution:** 3 good
**Rating:** 4
**Confidence:** 5

**Summary:**

Based on recent vision-language fundamental models such as GLIP and Grounding DINO, the authors propose an improved multimodal query pipeline. A Gated Class-scalable Perceiver is used to apply cross-attention for both the language and vision query inputs. A masking strategy for text tokens is further proposed to improve the generalization ability. The proposed method achieves strong performance on zero-shot LVIS and ODinW benchmarks.

**Strengths:**

- This paper first explores using both vision and language queries as inputs
- The proposed method achieves superior experimental results over previous methods
- The authors release the source code

**Weaknesses:**

- Overclaimed training efficiency, see Q1 in the Questions section. The pre-training time of GLIP should also be considered, which makes MQ-GLIP more time and data-intensive.
- Potential violation of zero-shot setting, see Q2 in the Questions section. The MQ-Det might improperly use exemplar images in a zero-shot setting, which is against the task definition.
- Subpar full-shot performance, see Q3 in the Questions section.

**Questions:**

1. I want to verify one point: the proposed method, such as MQ-GLIP, is trained from scratch, or finetuned from the initial weights from GLIP? According to #175-176: "we freeze the entire pre-trained detector and only train the newly-added gated class-scalable perceiver modules", I assume that MQ-GLIP is finetuned on the pre-trained GLIP model. Then I think the author overclaims their training efficiency. I think the pre-trained time of GLIP cannot be ignored. For example, in #232, it should be "MQ-GLIP-T adds extra 2% training time and extra 12% data usage of GLIP-T." rather than "only requires 2%". In my opinion, MQ-GLIP increases the training time and training data compared to GLIP, since MQ-GLIP brings an extra fine-tuning stage.

2. When evaluating the zero-shot setting such as zero-shot LVIS, does the MQ-Det requires the exemplar images for each class of the LVIS dataset? I guess the answer is 'yes' since the MQ-Det requires both the language and vision queries as inputs. But I think this way may violate the zero-shot setting. In the zero-shot setting, the LVIS classes are treated as unseen classes, and the exemplar images of the unseen classes are not allowed to be accessed. The model can only use the textual descriptions or text attributes associated with the unseen classes. So I think MQ-Det achieves strong zero-shot performance via a cheating inference evaluation. I think using the exemplar images of test-set classes is allowed for the few-shot setting but is not allowed for the zero-shot setting, which will weaken the contribution of this paper.

3. In table-2, the proposed MQ-Det performs worse than DINO-Swin-L on the full-shot setting. Can the authors provide some explainable reasons?

**Limitations:**

The authors discussed their limitations in section C of the appendix.

---

> ### Author Rebuttal · Authors · 2023-08-06
>
> We thank the reviewer for the feedback. Below, please find our responses to each of the concerns or questions raised in the review:
> > **Training efficiency on time and data**
>
> We acknowledge that our model is built upon pretrained GLIP/GroundingDINO models, which indirectly utilizes their pretraining datasets and accounts for more training time if trained from scratch. That is the reason we refer to our training process as modulated pretraining (lines 156-159) instead of pretraining.
>
> Meanwhile, we'd like to emphasize that the aforementioned issue does not contradict the efficiency of our method. Our efficiency lies in the fact that our approach allows current mainstream language-queried detectors to be equiped with multi-modal queries only through a lightweight modulating process. This offers a promising solution to overcome the limitations on insufficient granularity and ambiguous queries in current language-queried detectors. As an example, one may say that LoRA [1] (low-rank adaptation for efficient tuning on GPT) takes more training time and data than GPT if trained from scratch, but clearly, its efficiency lies in the fact that we do not need to train from scratch.
>
> We will make the following revision to avoid misleading:
>
> 1. We will provide more clarification on our data discrepancy in Table 1 and Table 2, for example:
>
> Table 1: xxxx. $^\dagger$ Modulating upon pretrained models indirectly utilizes their pre-training data, and potentially consumes more time if we take the training time of the pretrained language-queried detectors into consideration.
> | Model|...|Pre-train Data|...|Training Time|
> |-|-|-|-|-|
> |...|...|...|...|...|
> |MQ-GLIP-T|...|O365 (+GLIP$^\dagger$)|...|10$^\dagger$|
> |MQ-GroundingDINO-T|...|O365 (+GroundingDINO$^\dagger$)|...|10$^\dagger$|
>
> 2. Add more description on the  data discrepancy after the last sentence in line 232: "It is worth noting that modulating upon pretrained GLIP/GroundingDINO indirectly utilizes their pretraining data. The efficiency here describes that our approach allows current mainstream language-queried detectors to be equiped with multi-modal queries only through a lightweight modulating process, avoiding training from scratch."
>
> 3. Add clarification on the training time and data in lines 13-16,  lines 76-78, and lines 231-232. For example, "For instance, MQ-Det ... with merely additional 3% modulating time upon GLIP" in lines 13-16.
>
> [1] Hu, Edward J., et al. Lora: Low-rank adaptation of large language models. ICLR2022
> > **Zero-shot setting**
>
> We acknowledge that our setting is different from the zero-shot setting in previous language-queried detectors, since we use additional visual exemplars along with texts as category descriptions. Our setting is derived from practical implementation, where users can detect their customized objects through textual descriptions, visual exemplars, or both without any fine-tuning. As an example, detecting different types of "mushrooms" in the Mushroom dataset of OdinW-13 can be much easier with visual exemplars than ambiguous textual descriptions. It is hard to reach absolutely fair comparison in this setting despite a lot of efforts we have made. The reason is that most methods, except ours, do not support visual exemplars as inputs, while comparing our finetuning-free method with other finetuned methods in a few-shot setting would also be unfair. To avoid misleading, we will make the following revisions:
>
> 1. Modify the setting name:
>
>    (a) The name of section 3.2.1: "Multi-modal queried detection without finetuning"
>
>    (b) In the begining of line 226: "We evaluate the model's ability ... in a **finetuning-free setting**, where users can detect their customized objects through textual descriptions, visual exemplars, or both without any finetuning. "
>
> 2. Add more clarification:
>
>    (a) In the title of Table 1: "**Finetuning-free detection**  results on the LVIS benchmark. Differently, we provide models with 5-shot instances as vision queries without any fine-tuning..."
>
>    (b) Lines 13-15 in the abstract: "For instance, MQ-Det significantly improves the state-of-the-art open-set detector GLIP by +7.8% AP on the LVIS benchmark with multi-modal queries without any downstream fine-tuning..."
>
>    (c) Similar modifications as (b) in line 52, lines 76-77, and lines 226-232.
>
> Additionally, we provide two more methods to acquire vision queries in the **finetuning-free setting (the zero-shot setting in the original version)** in Section A.1 of the Appendix, which do not have access to the target dataset.
>
> We would also like to emphasize that the **core merit of this work** is that, we address the insufficient granularity and ambiguous queries that existing language-queried detectors suffer from via multi-modal queries, as recognized by Reviewer RdSN "The paper **tackles an important problem**: ..." and Reviewer JLFL "Leveraging both textual and visual info ... **makes lots of sense**". The main message from our original zero-shot comparison is that, equipped with multi-modal queries, previous language-queried detectors are allowed to detect objects with various  granularity and demonstrate great performance improvement without any finetuning. This confirms our core merit and indicates that multi-modal queried object detection can be a promising future direction.
> > **Full-shot performance**
>
> In the full-shot setting, the reasons that MQ-Det performs worse than DINO-Swin-L lie in twofold. First, our approach is more suitable in few-shot scenarios, where the auxiliary information provided by the vision queries plays a vital role. This information  takes weaker effect with sufficient data in the full-shot setting. Second, MQ-GLIP-T is modulated upon GLIP-T, with only the expectation  of improving over GLIP-T. Given that DINO-L is a larger model and holds stronger architecture design (e.g., mixed query selection and look-forward-twice module), it is understandable for MQ-GLIP-T to perform worse than it.

---

> > ### Comment · Reviewer_wDpN · 2023-08-18
> >
> > Thanks to the authors for providing the rebuttal replies.
> >
> > > Training efficiency on time and data
> >
> > The authors agree that it should be “add extra” rather than “only requires” in #232. In Table 1, the authors may consider adding two columns, e.g., ‘training time’ and 'modulating time’, ‘training data’ and ‘modulating data’ to avoid confusion.
> >
> > > Zero-shot setting
> >
> > While the authors acknowledge that their experiments do not conform to the zero-shot setting, they introduce a new term, the "finetuning-free setting". However, my concerns remain. For instance, the comparison in Table 1 is not fair because other methods did not use the 5 vision examples.  Also, since the authors propose a new setting, they may need more effort to provide more baselines rather than the variants of MQ-GLIP-T in Appendix A.1 under their new setting.
> >
> > Reviewer 9Uan also expressed concerns about this particular point. I also have the same concern with the reviewers 9Uan about missing the ablation studies about the number of vision examples used in the "finetuning-free setting".
> >
> > > Full-shot performance
> >
> > I am still confused about why the authors do not provide the MQ-GLIP-L results in the few-shot/full-shot setting.
> >
> > Overall, I will keep my initial rating.

---

> > > ### Author Response · Authors · 2023-08-19
> > >
> > > Thanks for the reviewer's response. Our new response is as follows.
> > >
> > >
> > >
> > > > Q1: In Table 1, the authors may consider adding two columns, e.g., ‘training time’ and 'modulating time’, ‘training data’ and ‘modulating data’ to avoid confusion.
> > >
> > > Thanks for the reviewer's valuable recommendation. We will revise the table according to your suggestion.
> > >
> > > > Q2: For instance, the comparison in Table 1 is not fair because other methods did not use the 5 vision examples. Also, since the authors propose a new setting, they may need more effort to provide more baselines rather than the variants of MQ-GLIP-T in Appendix A.1 under their new setting.
> > >
> > > 1. The finetuning-free comparison between our multi-modal queried method and previous language-queried methods is reasonable because it verifies the superiority of multi-modal queries over singe-modal queries, and shows that multi-modal queried object detection can be a promising future direction, rather than simply presenting a state-of-the-art model.
> > >
> > > 2. Thanks for the reviewer's suggestion. It is hard to find existing baselines that suitable in the multi-modal queried object detection setting, as recognized in the reviewer's initial review: "This paper first explores using both vision and language queries as inputs". To this end, we modify GLIP/GroundingDINO through the method in OWL-ViT as baselines and conduct finetuning-free evaluation in LVIS MiniVal.
> > >
> > >     Specifically, our modification contains three steps. 1) Acquiring naive vision quries: we feed the visual examplars togather with LVIS 1203 category texts into GLIP/GroundingDINO, then crop the corresponding regions on the output image features using an RoIPooler. The averaged cropped region features are treated as vision queries. 2) Constructing naive multi-modal queries: we average the classification logits of original language queries and the naive vision queries as the results of multi-modal queries.  3) Detection evaluation: we separately use the original language queries, the naive vision queries, and the naive multi-modal queries for evaluation on LVIS MiniVal. The classification is conducted via dot-product similarity that similar to OWL-ViT. The results are shown in the following table. For example , GLIP-T, GLIP-T-Img, and GLIP-T-MM denote GLIP with original language-queries, naive vision queries, and naive multi-modal queries, respectively. The number of vision queries is set to 5.
> > >
> > > | Model               | $AP$ | $AP_r$ | $AP_c$ | $AP_f$ |
> > > | ------------------- | ---- | ------ | ------ | ------ |
> > > | GroundingDINO-T     | 25.7 | 15.2   | 21.9   | 30.9   |
> > > | GroundingDINO-T-Img | 7.7  | 2.6    | 6.5    | 9.8    |
> > > | GroundingDINO-T-MM  | 14.7 | 8.2    | 12.6   | 17.9   |
> > > | MQ-GroundingDINO-T  | 30.2 | 21.7   | 26.2   | 35.2   |
> > > | GLIP-T              | 26.0 | 20.8   | 21.4   | 31.0   |
> > > | GLIP-T-Img          | 7.6  | 2.4    | 6.8    | 9.5    |
> > > | GLIP-T-MM           | 15.4 | 10.6   | 13.4   | 18.0   |
> > > | MQ-GLIP-T           | 30.4 | 21.0   | 27.5   | 34.6   |
> > > | GLIP-L              | 37.3 | 28.2   | 34.3   | 41.5   |
> > > | GLIP-L-Img          | 10.9 | 4.1    | 9.2    | 13.7   |
> > > | GLIP-L-MM           | 24.3 | 17.7   | 21.5   | 27.9   |
> > > | MQ-GLIP-L           | 43.4 | 34.5   | 41.2   | 46.9   |
> > >
> > > The results show that directly using vision queries in language-queried detectors through some naive modification demonstrates poor detection performance, and combining such vision queries with language queries as multi-modal queries impairs the performance.
> > >
> > >
> > > > Q3:  I also have the same concern with the reviewers 9Uan about missing the ablation studies about the number of vision examples used in the "finetuning-free setting".
> > >
> > > We have provided the ablation results in the rebuttal. Please refer to the `Meta-parameters and the related work chapter` part of our initial response to Reviewer 9Uan.
> > >
> > > > Q4: I am still confused about why the authors do not provide the MQ-GLIP-L results in the few-shot/full-shot setting.
> > >
> > > We have provided the results in the paper. Please refer to Figure 3 in the paper and Table Ⅲ in the Appendix. We did not present the MQ-GLIP-L results in  Table 2 only for fair comparison.
> > >
> > > ---
> > >
> > > Please feel free to let us know if you have other questions.

---

### Official Review · Reviewer_9Uan · 2023-07-06

**Soundness:** 3 good
**Presentation:** 4 excellent
**Contribution:** 3 good
**Rating:** 7
**Confidence:** 5

**Summary:**

The paper propose MQ-Det, a novel module that integrates both language and visual queries efficiently for object detection tasks. This module enhances each category token with vision queries, providing rich detailed visual context to the text-based models. The proposed method is experimentally tested on LVIS and ODinW datasets, obtaining state of the art results. The method is versatile and can be applied to other state of the art language queried object detectors.

**Strengths:**

The overall idea of the paper is easy to understand.
The presented method, MQ-Det is novel because it introduces a unique combination of language and visual queries (multi-modal queries) to enhance object detection in an efficient manner.
To train the proposed module it's not expensive and it does not require a lot of data.
The proposed method, MQ-Det, was evaluated using two state of the art models, GLIP and GroundingDINO. The results demonstrated superior performance when MQ-Det was applied, showcasing the effectiveness of the MQ-Det approach.

**Weaknesses:**

The comparison in the zero-shot and few-shot scenario is not fair since this method uses "5 instances as vision queries for each category from the downstream training set" thus having access to some information from the target dataset while all the other methods do not have any kind of access to that (for example owl-vit for image guided detection also doesn't use any fine-tuning, but when using vision queries from the target dataset they present the results as few-shot -- please see chapter 4.4 from owl-vit). Same observation for few-shot. Also it's not clear how are those 5 instances picked and how does that affect the final performance.

The time and data comparison against state of the art it's not entirely fair since this method builds on top of GLIP and GroundingDINO. So this method benefits of the training data and training time of GLIP/GroundingDINO, thus the affirmations and comparisons may be misleading.

It's not clear if the method can be used with text only or you always need some vision queries. Tied to that it's not clear what MQ-GLIP-T-Txt from Table 1 represents. Can you mask the visual queries? The comparison between MQ-GLIP-T-Txt and GLIP-T it's confusing.

Various meta-parameters seem to be chosen randomly: for example choosing 5 as the vision queries.

Ultimately, while I do see the merits of this work, I find the comparisons to be misleading and not entirely fair.

minor typos: visual detials


After rebuttal: The authors addressed my concerns properly and with the promised modifications I consider that this paper meets NeurIPS standards.

**Questions:**

How are the vision queries chosen? How does this affect the performance?
Choosing to have the related work chapter as the forth one it's unusual, is there any reason for that?
please see Weaknesses.

**Limitations:**

the paper discuss some of the limitations throughout the paper and does not address societal impact.

---

> ### Author Rebuttal · Authors · 2023-08-06
>
> We thank the reviewer for the feedback. Please find our responses below:
> > **Comparison in the zero-shot scenario**
>
> We acknowledge that our setting is different from the zero-shot setting in previous language-queried detectors, since we use additional visual exemplars along with texts as category descriptions. Our setting is derived from practical implementation, where users can detect their customized objects through textual descriptions, visual exemplars, or both without any fine-tuning. As an example, detecting different types of "mushrooms" in the Mushroom dataset of OdinW-13 can be much easier with visual exemplars than ambiguous textual descriptions. It is hard to reach absolutely fair comparison in this setting despite a lot of efforts we have made. The reason is that most methods, except ours, do not support visual exemplars as inputs, while comparing our finetuning-free method with other finetuned methods in a few-shot setting would also be unfair. To avoid misleading, we will make the following revisions:
>
> 1. Modify the setting name:
>
>    (a) The name of section 3.2.1: "Multi-modal queried detection without finetuning"
>
>    (b) In the begining of line 226: "We evaluate the model's ability ... in a **finetuning-free setting**, where users can detect their customized objects through textual descriptions, visual exemplars, or both without any fine-tuning. "
>
> 2. Add more clarification:
>
>    (a) In the title of Table 1: "**Finetuning-free detection**  results on the LVIS benchmark. Differently, we provide models with 5-shot instances as vision queries without any fine-tuning..."
>
>    (b) Lines 13-15 in the abstract: "For instance, MQ-Det significantly improves the state-of-the-art open-set detector GLIP by +7.8% AP on the LVIS benchmark with multi-modal queries without any downstream fine-tuning..."
>
>    (c) Similar modifications as (b) in line 52, lines 76-77, and lines 226-232.
>
> Additionally, we provide two more methods to acquire vision queries in the **finetuning-free setting (the zero-shot setting in the original version)** in Section A.1 of the Appendix, which do not have access to the target dataset.
>
> We would also like to emphasize that the our goal is to address the insufficient granularity and ambiguous queries that existing language-queried detectors suffer from. To achieve this goal, we propose to equip language-queried-only detectors with multi-modal queries. The main message from our original zero-shot comparison is that, equipped with multi-modal queries, previous language-queried detectors are allowed to detect objects with various  granularity and demonstrate great performance improvement without any finetuning. This confirms our goal and indicates that multi-modal queried object detection can be a promising future direction.
> > **Comparison in the few-shot scenario**
>
> One thing we'd like to clarify is that the few-shot comparison is absolutely fair. The vision queries are strictly selected from the few-shot datasets, as described in lines 217-218. For example, in the 3-shot setting, we only use the 3-shot training samples as the vision queries for each category, without any additional data.
> > **Comparison on training time and data**
>
> We will provide more clarification to avoid misleading. Due to the character limit, please refer to the "`training efficiency on time and data`" part of our response to Reviewer wDpN, for detailed modifications we will make during revision.
> > **Inference with masked vision queries**
>
> Our approach supports inference with mixed queries, namely, only augmenting a part of categories with multi-modal queries while leaving other categories with single-modal queries. For example, as shown in the table, MQ-GLIP-T-Mix only provides "birdfeeder" with both textual descriptions and visual exemplars, while providing "armchair" with only visual exemplars and "straw" with only textual descriptions. MQ-GLIP-T-Txt only uses text queries, which is equivalent to GLIP. We will add a clear definition on MQ-GLIP-T-Txt in line 238. MQ-GLIP-T-Img masks all input texts and only uses vision queries. The categories are from LVIS MiniVal and the AP results in the finetuning-free setting are reported.
> |Model|Armchair #19|Straw #1024|Birdfeeder #100|
> |-|-|-|-|
> | MQ-GLIP-T| 45.8| 12.0| 2.8|
> | MQ-GLIP-T-Txt | 44.1| 8.2| 0.0 |
> | MQ-GLIP-T-Img | 41.7| 12.3| 4.2 |
> | MQ-GLIP-T-Mix | 41.8 (Img) |8.9 (Txt)|2.8 (Txt+Img)|
>
> We did not include the mixed results in the initial submission because it is laborious to design a customized query type for each category in the massive benchmarks used in the paper. However, it is relatively easy for users to flexibly adjust the query types to meet their own needs during implementation. We leave this study to future work.
> > **How does the vision queries affect the final performance?**
>
> The 5 vision queries are randomly picked. We did not employ specific tricks for the selection of vision queries, as even the lower bound achieved through random sampling outperformed the language-queried baseline models. The detailed results with multiple random sampling can be found in the "`inference variance with error bars`" part of our response to Reviewer RdSN.
> > **Meta-parameters and the related work chapter**
>
> We did not conduct a specific search on meta-parameters since it is not the key point of this work. Nonetheless, here we provide an analysis of the number of vision queries in the finetuning-free setting. We observed a clear improvement in detection performance even with a minimal number of vision queries, e.g., 1 vision query.
>
> | #Vision queries|0|1|3|5|10|
> |-|-|-|-|-|-|
> |LVIS AP (Finetuning-free)|26.0|28.5|29.9|30.4|30.6|
>
> We present the related work chapter as the fourth one to encourage readers to focus on our method. A similar pattern can be found in [1], [2].
>
> [1] Radford A, et al. Learning transferable visual models from natural language supervision. ICML2021.
>
> [2] Chen T, et al. A unified sequence interface for vision tasks. NeurIPS2022.

---

> > ### Comment · Reviewer_9Uan · 2023-08-18
> >
> > I appreciate the authors' efforts in addressing the concerns through their rebuttal and the supplementary experiments. I recognize the merits of the proposed method and commend the clarity of the manuscript's presentation. However, my reservations persist regarding the fairness of the zero-shot comparisons. Despite the authors' acknowledgment of this issue, the forthcoming modifications may only partially alleviate the reader's uncertainties. It's pertinent to note that certain methods, even in a zero-shot paradigm without any fine-tuning or weight updates, might inherently benefit from the availability of 5 samples, as exemplified by the "owl-vit" method.

---

> > ### Comment · Reviewer_9Uan · 2023-08-18
> >
> > How are the results changing in the zero-shot setup if only one example is used as opposed to 5?

---

> > > ### Author Response · Authors · 2023-08-19
> > >
> > > Thanks for the reviewer's response. Our new response is as follows.
> > >
> > > > Q1: It's pertinent to note that certain methods, even in a zero-shot paradigm without any fine-tuning or weight updates, might inherently benefit from the availability of 5 samples, as exemplified by the "owl-vit" method.
> > >
> > > 1. In fact, "owl-vit" did not demonstrate the benefits from vision samples compared to its language-queried counterpart. In chapter 4.4 of the "owl-vit", only the results of using 1 vision sample and 10 vision samples were compared, without including its language-queried results. To verify this, we conducted experiments using its code in Huggingface and compared its language-queried model, OWL-ViT-Txt, with its vision-queried model, OWL-ViT-Img, as shown in the table below. OWL-ViT-Img, MQ-GLIP-T, and MQ-GLIP-T-Img all use one identical vision sample as the vision query. The results show that the usage of vision samples alone in OWL-ViT showcases inferior open-world detection performance. Furthermore, "owl-vit" does not support language and vision queries as joint inputs, which neglects the complementary nature of vision and language, as outlined in our related work chapter.
> > >
> > > | Model                  | Rabbit | Pothole | ODinW-13 |
> > > | -| ------ | ------- | -------- |
> > > | OWL-ViT-Txt (ViT-L/14) | 73.0   | 17.5    | 40.9     |
> > > | OWL-ViT-Img (ViT-L/14) | 19.4   | 0.3     | 10.5     |
> > > | MQ-GLIP-T | 74.4   | 13.2    | 43.9     |
> > > | MQ-GLIP-T-Txt          | 71.6   | 6.7     | 41.9     |
> > > | MQ-GLIP-T-Img          | 71.1   | 4.0     | 29.6     |
> > >
> > > 2. We are the first work to explore multi-modal queries in object detection, which supports jointly using textual descriptions and visual exemplars. The finetuning-free comparison between our multi-modal queried method and previous language-queried methods is reasonable because it verifies the superiority of multi-modal queries over singe-modal queries, and shows that multi-modal queried object detection can be a promising future direction, rather than simply presenting a state-of-the-art model.
> > >
> > > 3. It is challenging to use multi-modal queries without any finetuning for previous language-queried detectors. We conduct further experiments to show that, directly using vision queries in language-queried detectors through some naive modification demonstrates poor detection performance, and combining such vision queries with language queries as multi-modal queries impairs the performance.
> > >
> > >    We modify GLIP/GroundingDINO through the method in OWL-ViT as baselines and conduct finetuning-free evaluation on LVIS MiniVal. Specifically, our modification contains three steps. 1) Acquiring naive vision quries: we feed the visual examplars togather with LVIS 1203 category texts into GLIP/GroundingDINO, then crop the corresponding regions on the output image features using an RoIPooler. The averaged cropped region features are treated as vision queries. 2) Constructing naive multi-modal queries: we average the classification logits of original language queries and the naive vision queries as the results of multi-modal queries.  3) Detection evaluation: we separately use the original language queries, the naive vision queries, and the naive multi-modal queries for evaluation on LVIS MiniVal. The classification is conducted via dot-product similarity that is similar to OWL-ViT. The results are shown in the following table. For example , GLIP-T, GLIP-T-Img, and GLIP-T-MM denote GLIP with original language-queries, naive vision queries, and naive multi-modal queries, respectively. The number of vision queries is set to 5.
> > >
> > >
> > >
> > > | Model               | $AP$ | $AP_r$ | $AP_c$ | $AP_f$ |
> > > | -| ---- | ------ | ------ | ------ |
> > > | GroundingDINO-T     | 25.7 | 15.2   | 21.9   | 30.9   |
> > > | GroundingDINO-T-Img | 7.7  | 2.6    | 6.5    | 9.8    |
> > > | GroundingDINO-T-MM  | 14.7 | 8.2    | 12.6   | 17.9   |
> > > | MQ-GroundingDINO-T  | 30.2 | 21.7   | 26.2   | 35.2   |
> > > | GLIP-T              | 26.0 | 20.8   | 21.4   | 31.0   |
> > > | GLIP-T-Img          | 7.6  | 2.4    | 6.8    | 9.5    |
> > > | GLIP-T-MM           | 15.4 | 10.6   | 13.4   | 18.0   |
> > > | MQ-GLIP-T           | 30.4 | 21.0   | 27.5   | 34.6   |
> > > | GLIP-L              | 37.3 | 28.2   | 34.3   | 41.5   |
> > > | GLIP-L-Img          | 10.9 | 4.1    | 9.2    | 13.7   |
> > > | GLIP-L-MM           | 24.3 | 17.7   | 21.5   | 27.9   |
> > > | MQ-GLIP-L           | 43.4 | 34.5   | 41.2   | 46.9   |
> > >
> > >
> > >
> > > > Q2: How are the results changing in the zero-shot setup if only one example is used as opposed to 5?
> > >
> > > Please refer to the `Meta-parameters and the related work chapter` part of our initial response for the results of MQ-GLIP-T on LVIS MiniVal. Also, the MQ-GLIP-T and MQ-GLIP-T-Img in the first table uses one example for ODinW-13 evaluation in the  zero-shot setup, namely,
> > >
> > > | Model         | ODinW-13 (1 vision query) | ODinW-13 (5 vision queries) |
> > > | -| - | -|
> > > | MQ-GLIP-T     | 43.9   | 45.6   |
> > > | MQ-GLIP-T-Img | 29.6  | 31.9   |
> > >
> > > ---
> > >
> > >  If you have other concerns, please feel free to reply.

---

> > > > ### Comment · Reviewer_9Uan · 2023-08-19
> > > >
> > > > I appreciate the clarity and structure of the paper and find the proposed method to be innovative. My primary concern is ensuring the clarity of the zero-shot setup for readers. I've noted your intention to replace "zero-shot" with "finetuning-free," which I believe is a step in the right direction. Can you confirm that this terminology will be consistent throughout the manuscript? Furthermore, I suggest adding a column in Tab.1 that clearly indicates the number of examples each method utilizes without finetuning. This won't detract from the strengths of your paper, as it stands robustly on its own merits. I simply believe this will aid readers in grasping the setup more effortlessly. I'm inclined to enhance my review score if we can achieve consensus on this "finetuning-free" presentation aspect.

---

> > > > > ### Author Response · Authors · 2023-08-19
> > > > >
> > > > > We thank the reviewer's recognition and valuable suggestions. We will thoroughly revise the paper for the clarity of the "finetuning-free" presentation.
> > > > >
> > > > > > **We will ensure the consistency in the usage of the terminology "finetuning-free" throughout the manuscript. The detailed modifications that we will make are listed as follows.**
> > > > >
> > > > > Generally, we will add a clear defination on "finetuning-free" in the introduction and experiment chapters. We will also thoroughly revise the details in the paper as follows.
> > > > >
> > > > > ***Abstract*:**
> > > > >
> > > > > **Lines 13-16:** revise the sentence to "For instance, MQ-Det significantly improves the state-of-the-art open-set detector GLIP by +7.8% AP on the LVIS benchmark via multi-modal queries without any downstream fine-tuning, ... , with merely additional 3% modulating time upon GLIP."
> > > > >
> > > > > ***Introduction*:**
> > > > >
> > > > > **Lines 50-53:** add the definition of the "finetuning-free" setting and revise the sentence to "We evaluated our models on a finetuning-free setting, where users can detect their customized objects through textual descriptions, visual exemplars, or both without any fine-tuning. With only one epoch ... , our approach impressively improves the finetuning-free performance by +7.8% on the LVIS benchmark through providing the model with 5 visual exemplars along with textual category descriptions."
> > > > >
> > > > > **Lines 74-78:** revise the sentence to "Our proposed MQ-Det demonstrates powerful transferability on the finetuning-free and few-shot scenarios, ...  Specifically, MQ-Det outperforms GLIP by +7.8% AP in the finetuning-free detection on the challenging LVIS [13] and averagely +6.3% AP on 13 downstream few-shot detection tasks [23], modulated with merely 3% of the training time required by GLIP."
> > > > >
> > > > > ***Experiments*:**
> > > > >
> > > > > ***Chapter 3.1.2*:**
> > > > >
> > > > > **Lines 218-220:** highlight the definition of the "finetuning-free" setting and revise the sentence to "We also evaluate our method in a finetuning-free setting, namely, users can detect their customized objects through textual descriptions, visual exemplars, or both without any fine-tuning. During finetuning-free evaluation, we extract 5 instances as vision queries for each category from the downstream training set without any finetuning."
> > > > >
> > > > > ***Chapter 3.2.1*:**
> > > > >
> > > > > **Modify the chapter name**: "Multi-modal queried detection without finetuning".
> > > > >
> > > > > **The begining of line 226:** add the definition of the "finetuning-free" setting: "We evaluate the model's ability ... in a finetuning-free setting, where users can detect their customized objects through textual descriptions, visual exemplars, or both without any fine-tuning. "
> > > > >
> > > > > **Lines 228-231:** "Overall, MQ-Det demonstrates strong finetuning-free transferability with ... MQ-GLIP-L surpasses the current SoTA on LVIS by a large margin with simply 5 visual exemplars provided, reaching ..., which verifies the superiority of our multi-modal queries over single-modal queries. Meanwhile, MQ-Det demonstrates good training efficiency, e.g., MQ-GLIP-T only requires additional 2% training time and 12% data usage of GLIP-T when modulated upon GLIP-T. It is worth noting that modulating upon pretrained GLIP/GroundingDINO indirectly utilizes their pretraining data. The efficiency here describes that our approach allows current mainstream language-queried detectors to be equiped with multi-modal queries only through a lightweight modulating process, avoiding training from scratch."
> > > > >
> > > > > ***Conclusion*:**
> > > > >
> > > > > **Lines 343**: "MQ-Det shows promising gains on the finetuning-free and few-shot settings ..."
> > > > >
> > > > >
> > > > >
> > > > > ***Others*:**
> > > > >
> > > > > We will revise Table 1 and Table 2 based on the reviewer's suggestions. We will replace the "zero-shot" in the title of Table 3 with "finetuning-free".
> > > > >
> > > > > Similar revisions will be made in the appendix.
> > > > >
> > > > >
> > > > >
> > > > > > **We will modify Table 1 following the reviewer's suggestion.  For example:**
> > > > >
> > > > > Table 1: Finetuning-free detection results on the LVIS benchmark. Differently, we provide models with 5-shot instances as vision queries along with the language queries without any fine-tuning. ... $^\dagger$ Modulating upon pretrained models indirectly utilizes their pre-training data, and consumes more time if we take the training time of the pretrained language-queried detectors into consideration.
> > > > >
> > > > > | Model| ...  | # Vision Queries | ...  | Pre-train Data| ...  | Training Time | ...  |
> > > > > | -| -| -|-|-|-|-|-|
> > > > > | ...                | ...  | ...  | ...  | ...  | ...  | ...           | ...  |
> > > > > | GLIP-T             | ...  | 0                | ...  | O365, GoldG, Cap4M              |      | 480           | ...  |
> > > > > | ...                | ...  | ...   | ...  | ...                             | ...  | ...           | ...  |
> > > > > | MQ-GLIP-T          | ...  | 5                | ...  | O365 (+GLIP$^\dagger$)          | ...  | 10$^\dagger$  | ...  |
> > > > > | MQ-GroundingDINO-T | ...  | 5                | ..   | O365 (+GroundingDINO$^\dagger$) | ...  | 10$^\dagger$  | ...  |
> > > > >
> > > > > ---
> > > > >
> > > > > Please feel free to reply if you have further suggestions or questions.

---

> > > > > > ### Comment · Reviewer_9Uan · 2023-08-19
> > > > > >
> > > > > > I appreciate the proposed changes and believe they will enhance the clarity of the paper. I've revised my score to 'Accept'.

---

> > > > > > > ### Author Response · Authors · 2023-08-20
> > > > > > >
> > > > > > > Thanks for the reviewer's recognition and raising your score. Your suggestions significantly help improve the quality of our paper.

---

### Official Review · Reviewer_RdSN · 2023-07-06

**Soundness:** 4 excellent
**Presentation:** 3 good
**Contribution:** 3 good
**Rating:** 7
**Confidence:** 5

**Summary:**

The paper introduces MQ-Det, a novel approach for open-vocabulary object detection that combines textual descriptions and visual exemplars as category queries. MQ-Det aims to address the limitations of existing text-queried object detectors by incorporating visual information and providing various granularity in the descriptions. The proposed architecture can be easily integrated with pre-trained language-queried detectors. To overcome the learning inertia problem caused by frozen detectors, MQ-Det employs a vision-conditioned masked language prediction strategy. This strategy involves randomly masking text tokens and allowing vision queries to independently predict objects, ensuring sufficient visual intervention during training. Overall, MQ-Det presents a simple yet effective architecture and training strategy that improves open-world object detection performance by incorporating multi-modal queries.

**Strengths:**

1. The paper is well-written and easy to follow: The authors present their research in a clear and organized manner.
2. The paper tackles an important problem: The authors address the challenge of improving the robustness and generalization of current language-query based object detectors by incorporating visual cues. This is a significant problem in the field of open-vocabulary  detection, as relying solely on textual descriptions can lead to insufficient granularity and ambiguous queries. By introducing multi-modal queries, the paper provides a promising solution to overcome these limitations and enhance detection performance in real-world scenarios.
3.The proposed GCP module and training techniques have broader applicability: GCP module introduced in the paper is a valuable contribution that can have implications beyond the scope of this study. For example, conditional gating layer of the GCP module provide a framework for effectively integrating useful visual information (instead of noisy visual templates) into language-queried detectors. Additionally, the random masking strategy addresses the learning inertia problem and enables better fusion of visual and textual cues. These findings can benefit future research.
4. The experiment is solid and proves the effectivenss of the proposed method.

**Weaknesses:**

1. The training data in Table 1 and Table 2 is confusing. It appears that the authors loaded the pretrained GLIP models, which were pretrained on datasets such as Objects365, GoldG, CC4M, and Cap24M. However, the presented training data in those tables only includes the Objects365 dataset for proposed models. It is important to note that the performance of the model is achieved by utilizing all the mentioned pretraining datasets, rather than just the Objects365 dataset. Clarifying this discrepancy in the presentation of training data would provide a clearer understanding of the model's training process.

2. The evaluation of open-vocabulary or open-set object detection requires a clear separation of base and novel classes. Unfortunately, the paper does not explicitly evaluate the performance on these two sets separately. This omission makes it difficult to assess the model's generalization capability specifically on novel classes. To gain a deeper and better understanding of the model's performance in generating novel classes, it is recommended to conduct separate evaluations and present the results accordingly.

**Questions:**

As mentioned in the paper, the selection of visual templates during inference plays a critical role in the success of the proposed approach. I would like to inquire about the impact of random sampling templates on the performance. Specifically, I am interested in whether the authors performed multiple inference runs, averaged the performance results, and presented the inference variance with error bars.

Could you please clarify whether this analysis was conducted and whether the paper provides information on the effect of random sampling templates on performance?

**Limitations:**

I do not see potential negative societal impact.

---

> ### Author Rebuttal · Authors · 2023-08-06
>
> We thank the reviewer for the thoughtful feedback. Please find our responses below:
>
> > **Training data**
>
> Thank the reviewer for your valuable reminder. Our model is indeed  built upon pretrained GLIP/GroundingDINO models, which indirectly utilizes their pretraining datasets. We will provide the following clarification on the data discrepancy to avoid misleading.
>
> 1. We will clarify the data discrepancy in Table 1 and Table 2, for example:
>
> Table 1: xxxx. $^\dagger$ Modulating upon pretrained models indirectly utilizes their pre-training data, and potentially consumes more time if we take the training time of the pretrained language-queried detectors into consideration.
>
> | Model|...|Pre-train Data|...|Training Time|
> |-|-|-|-|-|
> |...|...|...|...|...|
> |MQ-GLIP-T|...|O365 (+GLIP$^\dagger$)|...|10$^\dagger$|
> |MQ-GroundingDINO-T|...|O365 (+GroundingDINO$^\dagger$)|...|10$^\dagger$|
>
> 2. Add more description on the  data discrepancy after the last sentence in line 232: "It is worth noting that modulating upon pretrained GLIP/GroundingDINO indirectly utilizes their pretraining data. The efficiency here describes that our approach allows current mainstream language-queried detectors to be equiped with multi-modal queries only through a lightweight modulating process, avoiding training from scratch."
>
> > **Explicit evaluation on an open-vocabulary detection setting**
>
> Thank the reviewer for the constructive recommendation. We add the following experiment to further investigate the model's generalization performance.
>
> We first construct a novel category set from 1,203 LVIS categories. Specifically, we remove the LVIS categories that exist in the 365 classes of Objects365 and finally obtain 986 novel categories that did not appear during our modulated pretraining. The remaining 217 categories are represented as base categories. Then, we conduct zero-shot inference on the separated categories to verify the generalization of multi-modal query learning. The results are shown in the following table.
>
> More details: We consider a category in LVIS as a base class if its name or synonyms appear in the category name set of Objects365. The names are all in lowercase and singular form, with all "()" and "_" removed.
>
> | Model              | $AP_{novel}$ | $AP_{base}$ | $AP_{all}$ |
> | ------------------ | ------------ | ----------- | ---------- |
> | GroundingDINO-T    | 22.1         | 36.7        | 25.6       |
> | GLIP-T             | 20.8         | 42.0        | 26.0       |
> | GLIP-L             | 35.4         | 45.5        | 37.9       |
> | MQ-GroundingDINO-T | 26.2         | 43.0        | 30.2       |
> | MQ-GLIP-T          | 26.5         | 42.8        | 30.4       |
> | MQ-GLIP-L          | 41.7         | 51.3        | 44.0       |
>
> The results indicate that multi-modal queries generalize well to novel classes that do not exist in the modulated pretraining. Specifically, +4.1%, +5.7%, and +6.3% AP on novel classes of MQ-GroundingDINO-T, MQ-GLIP-T, and MQ-GLIP-L over their baselines, respectively. We will include this experiment in the revised version.
>
> It is worth noting that the separation of base and novel classes differs from previous works on open-vocabulary detection (OVD). The reason is that the testing categories of previous separation are partially included in our training dataset Objects365. Therefore, we represent the classes in LVIS that do not exist in our modulated pretraining dataset Objects365 as novel classes. The frequency distribution of the separated LVIS dataset is shown in the following table:
>
> | Classes | #Rare | #Common | #Frequent |
> | ------- | ----- | ------- | --------- |
> | Novel   | 326   | 404     | 256       |
> | Base    | 11    | 57      | 149       |
> | All     | 337   | 461     | 405       |
>
> > **Inference variance with error bars**
>
> We did not report the averaged results of multiple runs in the initial submission. Here, we provide the averaged results with error bars from 3 inference runs using seeds of 3, 30, and 300 for both zero-shot and few-shot settings. Models with different seeds randomly select diffrent vision queries. We will include them in the revised version. Meanwhile, this work uses random sampling templates to investigate the feasibility of multi-modal queries. We leave the study of more complicated template sampling to future work.
>
> LVIS MiniVal zero-shot:
>
> | Model              | $AP$          | $AP_r$        | $AP_c$        | $AP_f$        |
> | ------------------ | ------------- | ------------- | ------------- | ------------- |
> | MQ-GLIP-T          | 30.5 $\pm$0.1 | 21.8 $\pm$0.8 | 27.4 $\pm$0.1 | 34.8 $\pm$0.2 |
> | MQ-GLIP-L          | 43.7$\pm$0.3  | 34.8 $\pm$0.3 | 41.6 $\pm$0.4 | 47.2 $\pm$0.3 |
> | MQ-GroundingDINO-T | 30.4 $\pm$0.2 | 21.8 $\pm$0.3 | 26.3 $\pm$0.1 | 35.4 $\pm$0.2 |
>
> OdinW zero-shot:
>
> | Model              | OdinW-35 $AP_{avg}$ | OdinW-13 $AP_{avg}$ |
> | ------------------ | ------------------- | ------------------- |
> | MQ-GLIP-T          | 20.7 $\pm$0.5       | 45.4 $\pm$0.6       |
> | MQ-GLIP-L          | 24.0 $\pm$0.4       | 54.1 $\pm$0.3       |
> | MQ-GroundingDINO-T | 22.5 $\pm$0.3       | 51.0 $\pm$0.3       |
>
> OdinW few-shot (3-shot):
>
> | Model     | OdinW-35 $AP_{avg}$ | OdinW-13 $AP_{avg}$ |
> | --------- | ------------------- | ------------------- |
> | MQ-GLIP-T | 43.1 $\pm$0.4       | 57.2 $\pm$0.5       |

---

> > ### Comment · Reviewer_RdSN · 2023-08-18
> >
> > Thanks, authors for the detailed response. It largely solves my concern. Thus, I keep my original rating in the current stage.

---

> > > ### Author Response · Authors · 2023-08-19
> > >
> > > Thanks for the reviewer's recognition and constructive suggestions.

---

### Official Review · Reviewer_JLFL · 2023-07-07

**Soundness:** 3 good
**Presentation:** 3 good
**Contribution:** 3 good
**Rating:** 6
**Confidence:** 4

**Summary:**

In this paper, the authors proposed MQ-Det which leverage both textual and visual information for object detection in the wild. The proposed plug-and-play GCP module is very compatible with existing mainstream architectures. The authors conducted extensive experiments on multiple benchmark datasets and showed improved performance over several baseline methods for both zero-shot and few-shot detection.

**Strengths:**

1. Leveraging both textual and visual info for open-vocabulary detection makes lots of sense as they provide different level of signals for recognizing objects as mentioned in the draft. The proposed GCP module and masking based training strategy is a novel technique to combine those informations.
2. The authors conducted extensive experiments and ablation studies to sufficiently validate the proposed approach over multiple baseline methods on several benchmarking datasets.
3. Writing is good and easy to follow. The tables and figures are also easy to understand.

**Weaknesses:**

1. For the last row of Tab. 3 (b) where the joint input leads to lower performance, the authors' explanation is that "this task may introduce redundant information and rise the learning difficulty." This is not very clear to me since redundancy could also lead to easier training rather than difficulty. Could you please elaborate more on this?
2. For Fig. 4, it is weird that the baseline 'No Vision Query' (blue box) model actually recognize the two zebras as either elk or horned cow. These two zebras looks very easy to recognize even without visual cue for large VL models. Is this due to insufficient training of baseline models or hand-picked examples to illustrate the idea?
3. As illustrated in Fig. 3, with increasing amount of training data, the gap between MQ-Det and other baseline methods are getting more and more closer. What would be a rule of thumb strategy for using MQ-Det versus other simpler methods?

**Questions:**

Please refer to the weaknesses section for details.

---

> ### Author Rebuttal · Authors · 2023-08-06
>
> We thank the reviewer for the feedback. Please find below our responses to the questions raised in the review:
>
> > **More explanation on redundancy with joint input in the last row of Tab. 3 (b)**
>
> The learning difficulty derives from the simple concatenated input $cat(\hat v_{i}, t)$, which equally considers vision and language information to learn the scale of vision queries. This may impede the gate on learning from vision information that contributes the most to the vision scales. In the future, we will explore a more reasonable combination of the two modalities to benefit the performance.
>
> > **Question on the baseline 'No Vision Query' model in Fig. 4**
>
> This example is selected from the bad cases of GLIP that is equal to the 'No Vision Query' model. The reasons of this failure are twofold. First, it is rather challenging to recognize "zebra" from all 1,203 categories in LVIS, since the model is more prone to make mispredictions as the number of categories increases. Second, "zebra" is not included in the detection training set (Objects365), making it a novel class. Through the use of multi-modal queries, we can provide the model with more clues to the category, thereby improving detection performance.
>
> > **A rule of thumb strategy for using MQ-Det versus other simpler methods**
>
> Thank the reviewer for the thoughtful question. We delve deeper into investigating whether there exists a rule of thumb strategy for deploying MQ-Det. In the table below, we gradually increase the data size (number of shots) and record the performance gap between MQ-GLIP-T (multi-modal queries) and GLIP-T (language queries). We report the averaged results on 13 benchmarks of OdinW-13 and two specific datasets.
>
> | Performance Gap                         | 0    | 1    | 3    | 5    | 10   | 50   | full |
> | --------------------------------------- | ---- | ---- | ---- | ---- | ---- | ---- | ---- |
> | $AP_{MQ-GLIP-T}-AP_{GLIP-T}$ (OdinW-13) | 3.7  | 5.8  | 6.3  | 4.7  | 3.5  | 1.1  | 0.6  |
> | $AP_{MQ-GLIP-T}-AP_{GLIP-T}$ (Aquarium) | 2.2  | 3.2  | 2.6  | 2.4  | 1.5  | 1.4  | 0.9  |
> | $AP_{MQ-GLIP-T}-AP_{GLIP-T}$ (Pothole)  | 6.7  | 5.8  | 6.2  | 7.1  | 7.3  | 2.3  | 0.2  |
>
> Here, we provide several points for the implementation guidance of MQ-Det:
>
> - A rule of thumb for using MQ-Det is that: MQ-Det is widely suitable for low-shot scenarios (empirically below 50-shot), but the case of the biggest improvement varies based on specific datasets. As exemplified in the table above, in OdinW-13, there generally exists an obvious gap between multi-modal queries and language queries under 10 shots. For a relatively simple task like Aquarium, the gap narrows significantly with just 10-shot learning. However, for a more challenging task like Pothole, which is to detect the potholes on the ground, the gap remains large even with 50-shot.
> - It is worth noting that employing MQ-Det is not laborious. The reason lies in twofold. First, our modulating process is efficient for training. Second, the GCP module and the masking strategy are easy to reproduce. We will open-source the complete code and provide more detailed guidance on how to incorporate MQ-Det into customized language-queried detectors, hoping that our work could embrace wider application.

---

> > ### Comment · Reviewer_JLFL · 2023-08-21
> >
> > Thanks for the responses. My main concerns are addressed and thus keep my original positive rating.

---

### Author Rebuttal · Authors · 2023-08-10

Dear reviewers, area chairs, senior area chairs, and program chairs,


We sincerely thank for the valuable comments. It is pleasure that this work has been fully recognized by Reviewer RdSN (Accept) and Reviewer JLFL (Weak Accept), including "**a novel approach**", "**a simple yet effective architecture and training strategy**", "**The paper tackles an important problem**", and "**makes lots of sense**". The main concerns of the other two reviewers lie in the fairness of the zero-shot comparison and the unclear statement of the data usage. In this regard, we explain in detail that our setting is a bit different from previous zero-shot setting and derived from practical implementation, where users can use multi-modal queries (textual descriptions, visual exemplars, or both) to detect a wider range of objects without any finetuning. Due to the setting gap, we have tried our best to conduct fair comparison. We also add detailed clarification on our data usage and our modulated training process to avoid misleading. We look forward to a better appreciation of this manuscript that incorporates our great efforts. Furthermore, this manuscript has been carefully revised according to the suggestions of the reviewers. We have made our code open-sourced, hoping to promote the development of open-world object detection.

The followings are our detailed responses. We greatly thank the constructive suggestions that significantly help improve the quality of our paper.

---

### Decision · Program_Chairs · 2023-09-21

**Decision:**

Accept (poster)

**Comment:**

After rebuttal and discussion three reviewers argue for acceptance, one for (borderline) rejection. The main points of criticism revolve around "overclaiming training efficiency" and a non-standard definition of the zero-shot setting. The authors acknowledge these shortcomings, and propose concrete changes to the paper. This is sufficient for an acceptance. The AC recommends the authors to implement these changes, as similar sentiments will arise in potential readers.